# Signature of quantum criticality in cuprates by charge density fluctuations

Riccardo Arpaia [1] ✉, Leonardo Martinelli[2], Marco Moretti Sala [2], Sergio Caprara [3,4], Abhishek Nag [5], Nicholas B. Brookes [6], Pietro Camisa[2], Qizhi Li[7], Qiang Gao [8], Xingjiang Zhou [8], Mirian Garcia-Fernandez [5], Ke-Jin Zhou [5], Enrico Schierle [9], Thilo Bauch [1], Ying Ying Peng [7], Carlo Di Castro [3], Marco Grilli [3,4], Floriana Lombardi [1], Lucio Braicovich [2,6] & Giacomo Ghiringhelli [2,10] ✉

The universality of the strange metal phase in many quantum materials is often attributed to the presence of a quantum critical point (QCP), a zero-temperature phase transition ruled by quantum fluctuations. In cuprates, where superconductivity hinders direct QCP observation, indirect evidence comes from the identification of fluctuations compatible with the strange metal phase. Here we show that the recently discovered charge density fluctuations (CDF) possess the right properties to be associated to a quantum phase transition. Using resonant x-ray scattering, we studied the CDF in two families of cuprate superconductors across a wide doping range (up to $p = 0.22$). At $p^* \approx 0.19$, the putative QCP, the CDF intensity peaks, and the characteristic energy $\Delta$ is minimum, marking a wedge-shaped region in the phase diagram indicative of a quantum critical behavior, albeit with anomalies. These findings strengthen the role of charge order in explaining strange metal phenomenology and provide insights into high-temperature superconductivity.

The strange metal phase in cuprate superconductors extends over a large portion of the doping-temperature phase diagram and is particularly robust at the doping level $p^*$, around which the maximum superconducting critical temperature is achieved[1–5]. At this doping, the resistivity is linear in $T$ down to the superconducting critical temperature $T_c$ or, in the presence of strong magnetic fields, to the lowest temperatures[3], electronic excitations lose their quasiparticle character[6], the Drude peak in the optical conductivity acquires an anomalous power-law decay in frequency[7], the magnetoresistance is linear in field[8] and the spin relaxation rate is almost $T$-independent[9].

The Fermi-liquid theory, although successful for strongly correlated normal metals, does not provide a proper description of this phenomenology. The unconventional properties of the strange metal are likely the result of strong electron correlation in a quasi-2D structure that leads to a complex energetic landscape where several electronic phases compete and may coexist. Indeed, a novel state has been recently proposed, with very strong long-range quantum entanglement and a thermalization time defined by the Planckian scattering time[10]. Other theories suggest that quasiparticles get spoiled by strong scattering either on nearly local excitations (such as the

[1]Quantum Device Physics Laboratory, Department of Microtechnology and Nanoscience, Chalmers University of Technology, SE-41296 Göteborg, Sweden. [2]Dipartimento di Fisica, Politecnico di Milano, Piazza Leonardo da Vinci 32, I-20133 Milano, Italy. [3]Dipartimento di Fisica, Università di Roma "La Sapienza", P.le Aldo Moro 5, I-00185 Roma, Italy. [4]CNR-ISC, via dei Taurini 19, I-00185 Roma, Italy. [5]Diamond Light Source, Harwell Campus, Didcot OX11 0DE, United Kingdom. [6]ESRF, The European Synchrotron, 71 Avenue des Martyrs, F-38000 Grenoble, France. [7]International Center for Quantum Materials, School of Physics, Peking University, CN-100871 Beijing, China. [8]Beijing National Laboratory for Condensed Matter Physics, Institute of Physics, Chinese Academy of Sciences, CN-100190 Beijing, China. [9]Helmholtz-Zentrum Berlin für Materialien und Energie, Albert-Einstein-Straße 15, D-12489 Berlin, Germany. [10]CNR-SPIN, Dipartimento di Fisica, Politecnico di Milano, Piazza Leonardo da Vinci 32, I-20133 Milano, Italy. ✉e-mail: riccardo.arpaia@chalmers.se; giacomo.ghiringhelli@polimi.it

pseudospin degrees of freedom of the Sachdev-Ye-Kitaev model[11,12], the charge density fluctuations[13], the short ranged antiferromagnetic fluctuations[14]) or on long-wavelength fluctuations (such as loop currents[15], phase fluctuations of incommensurate charge order[16]). In all cases, the physics of the strange metal phase seems compatible with the existence of a quantum critical point at the doping level $p^*$ (hereafter named QCP*)[15,17], although with an unconventional phenomenology[18]. In a hypothetical QCP scenario, both thermal and quantum fluctuations determine transport at high temperature but only the latter are relevant at low temperatures: with $p$ approaching $p^*$, one should then observe a vanishing characteristic energy $\Delta$[19]. In other words, $\Delta$ would set the doping-dependent temperature scale above which the strange metal region appears as a strong manifestation of a quantum critical behavior. The QCP* scenario would be confirmed by the recognition of the associated fluctuations and by the determination of their energy following the expected doping dependence. The experimental evidence of these two aspects has been so far elusive. Moreover, it is unclear whether they are of charge[20], spin[4,21], or mixed[22] nature. We note that although the energy scale of the pseudogap[23] is comparable to that of these fluctuations, the latter is not commonly considered to be correlated to the former[18,24,25].

Charge density fluctuations (CDF)[26], precursors of charge density waves[27–31] (CDW), were recently observed in several cuprate families[32], and pervade the whole strange metal region above and below the pseudogap temperature $T^*$. For their characteristics - presence in a broad range of doping in the phase diagram, finite energy and short correlation length that implies a broad almost isotropic $q$-space distribution - they were immediately connected to the strange metal phase of the cuprates. Besides the mere phenomenology, various theories put the CDF at the origin of the strange metal phenomenon, either by extending perturbatively into the strange metal region the approach of a Fermi liquid[13] or by waiving the concept of quasiparticle in favor of a quantum entangled matter[10]. Are CDF the quantum fluctuations at the origin of the QCP*?

To answer this question, we have used resonant x-ray scattering to measure the doping and temperature dependence of the energy, intensity and correlation length of CDF in YBa$_2$Cu$_3$O$_{7-\delta}$ (YBCO) and Bi$_2$Sr$_2$CaCu$_2$O$_{8+\delta}$ (Bi2212) samples. We spanned a broad doping range, with special attention for three cases, where the absence of CDW permits a thorough study of CDF, i.e., outside the quasi-static charge density wave dome ranging, e.g. in YBCO, from $p \sim 0.08$ to $p \sim 0.16$[27,31]. We thus chose $p = 0.22$, in the overdoped region, where the strange metal at high temperature is substituted by a the Fermi liquid behavior at lower temperatures; $p = 0.19 \approx p^*$ for both YBCO and Bi2212[33], where the strange metal extends down to the superconducting critical temperature $T_c$; and $p = 0.06$, where the strange metal behavior is observed only at relatively high temperatures[33,34]. High resolution spectra at the CDF critical wave vector $\mathbf{q}_{CDF}$ provide a direct measurement of the doping dependence of the characteristic energy parameter $\Delta(T,p)$, which is minimum at $p^*$ and, at fixed $p$, grows with temperature. Moreover, the quasi-elastic spectral weight increases almost constantly with temperature, everywhere in the reciprocal space (see Supplementary Figs. 1a, b, 2a, b, and 3a). Such isotropic rise of scattered intensity was disregarded in previous experiments. On the contrary, here we carefully analyze this phenomenon and find that the quasi-elastic signal grows linearly at high temperatures, more steeply at $p^*$ than in underdoped samples, following the Bose distribution function in the semiclassical regime (see Supplementary Fig. 4a). Thus, far from $\mathbf{q}_{CDF}$, we can associate a $T$-independent energy $\Omega$ that follows the same doping dependence of $\Delta$ (the latter representing the characteristic energy of the fluctuations at $\mathbf{q} = \mathbf{q}_{CDF}$ and at the lowest measured temperature). All this experimental evidence suggests the presence of a link between CDF and the QCP*.

# Results

## CDF at $p = 0.19$

We have performed resonant inelastic x-ray scattering (RIXS) and energy-integrated resonant x-ray scattering (EI-RXS) at the Cu $L_3$ edge (~930 eV) at different momentum transfer components $q$ parallel to the CuO$_2$ planes along the ($H$,0) and ($H$,$H$) directions, on slightly overdoped ($p \sim 0.19$) samples of YBCO and Bi2212 (see Methods for details about the sample growth). Figure 1 summarizes the very-high-resolution (~38 meV) RIXS results at 80 K and 200 K, i.e., close to and well above $T_c$, respectively. To properly map the CDF signal, panels 1a and 1b represent as colormaps the difference between the spectra measured along the ($H$,0) and ($H$,$H$) directions. In this way the elastic scattering signal originated from surface defects gets mostly canceled out and the weak CDF signal emerges at small but finite energy, with broad intensity peaks at $q = q_{CDF} \sim (0.30,0)$ for YBCO and $q = q_{CDF} \sim (0.25,0)$ for Bi2212. Each spectrum of the ($H$,0) series can be fitted below 150 meV with four narrow Gaussian peaks (elastic, CDF, bond-stretching phonons, phonon overtones in increasing energy loss), plus a broader feature for the spin and particle-hole excitations (see Methods, and Supplementary Fig. 5). As shown in panels 1d-i, the elastic peak intensity is temperature independent, with a monotonic decrease vs $H$ as in undoped samples (see Supplementary Fig. 6a) due to the absence of contribution from the CDW, as expected at doping levels above optimum[27,31]. On the contrary, the CDF intensity, at finite energy, is $T$-dependent and broad both in energy and in momentum. The FWHM of the CDF Gaussian peak, 55–60 meV, is significantly larger than the experimental energy resolution, and the energy position increases from $\Delta$ ~10 meV at 80 K to ~15 meV at 200 K. The CDF intensity plotted versus $q$ is peaked at $H = H_{CDF}$ with a FWHM ~ 0.15 r.l.u., corresponding to a rather short-ranged correlation length. The shape of this broad-in-$q$ peak is very similar at the two temperatures, as already observed in ref. 26, but its intensity is almost isotropically stronger at 200 K than at 80 K, for reasons discussed below. Finally, panels 1 f,i show the softening of the bond-stretching phonon, with an energy drop $\Delta E_{ph}$, measured at a $H$ value close but higher than $H_{CDF}$, which is larger at 80 K ($\Delta E_{ph}$ ~ 15 meV) than at 200 K. This indicates that, as opposite to the most common interpretations[35–39], the phonon softening is not associated only to long-ranged CDW but also, or mainly, to CDF.

## Temperature dependence of CDF

For an extensive analysis of the temperature evolution of the CDF signal at $p \sim 0.19$, we have used medium resolution RIXS (~62 meV, see Fig. 2a) and EI-RXS data, which are quicker to measure but do not allow the direct determination of the characteristic energy $\Delta$, because the CDF contribution cannot be resolved from the elastic one (see Methods and Supplementary Fig. 7). In this case, to best single out the CDF component and minimize the contribution of the bond-stretching phonons, in Fig. 2 we plot, as function of momentum, the integral of the RIXS spectra up to 35 meV energy loss (vertical band in Fig. 2a). This integral intensity is dominated by the CDF on top of other elastic and quasi-elastic components. A peak is present along the ($H$,0) direction but not along ($H$,$H$) (Fig. 2b, c, Supplementary Figs. 1a, b and 2a, b), consistently with the high resolution RIXS data and with ref. 26. Moreover, along both directions this quasi-elastic intensity increases with the temperature. It is evident that the $T$ and $q$ dependence of the CDF scattering intensity cannot be properly described by a single, temperature independent energy parameter. Therefore, we have modeled the CDF component $I_{CDF}(q,\omega)$ of the RIXS spectra according to the theory of the charge density instability in a highly-correlated Fermi liquid[40], which is particularly appropriate for the slightly overdoped samples. The theoretical x-ray scattering intensity is given by

$$I_{CDF}(\mathbf{q}, \omega) \propto Im[D(\mathbf{q}, \omega)] \cdot b(\omega), \tag{1}$$

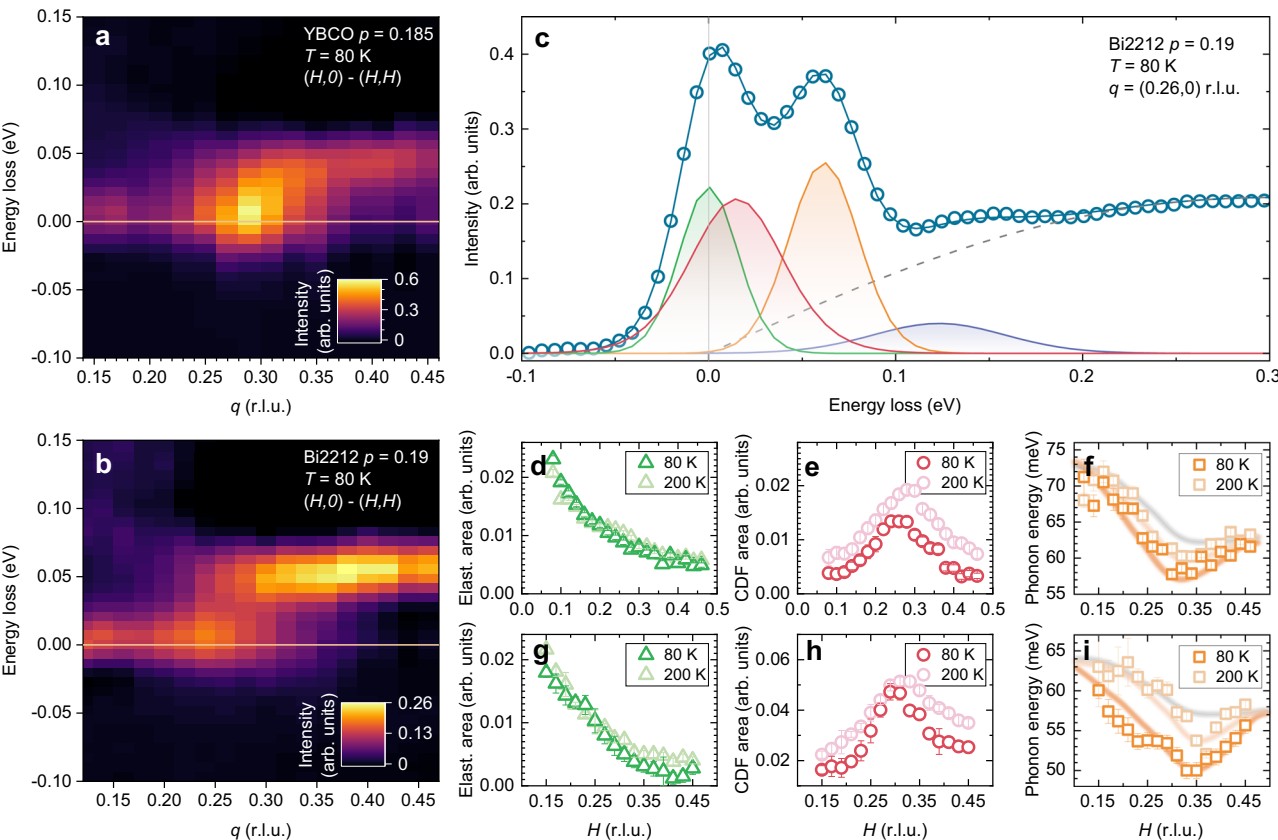

**Fig. 1 | Charge density fluctuations in overdoped cuprates.** High resolution ($\Delta E = 38$ meV) RIXS spectra have been measured on YBCO and Bi2212 ($p \approx 0.19$) at several momenta along both the $(H,0)$ and $(H,H)$ directions, at $T = 80$ K and $T = 200$ K. **a, b** Intensity maps of the difference $(H,0) - (H,H)$ taken at 80 K on YBCO and Bi2212. **c** Fit of a RIXS spectrum on Bi2212 at a representative momentum. The green, red, orange, blue Gaussians and the region below the gray dashed line represent respectively the pure elastic (mainly given by the specular peak centered at $\Gamma = (0,0)$), the CDF, the bond-stretching phonon modes, the bond-stretching overtone and the paramagnons. Additional details on the fit are provided in the Methods section. Given the position, intensity and width of the Gaussians, we have obtained, as a function of $q$ along the $(H,0)$ direction and at both temperatures, **d** the area of the elastic line, **e** the area of the CDF peak, and **f** the bond-stretching phonon dispersion. The error bars are estimated using the 95% confidence interval of the fit. In panel **f** the orange lines are guides to the eye while the gray line represents the phonon dispersion in absence of any softening, as measured in ref. 69. **g–i** Same as **d–f**, but on YBCO. In panel **i**, the gray line represents the phonon dispersion in absence of any softening, as measured in ref. 70.

i.e., the product of the $T$-dependent Bose distribution $b(\omega) = (1 - e^{-\omega/k_B T})^{-1}$ with the imaginary part of the dynamical density fluctuation propagator $D(q,\omega)$, where the $\hbar$ constant is implicit so that the $\omega$ terms stand for energy, and $\omega > 0$ for energy loss (Stokes) scattering. The propagator is that of overdamped quantum critical fluctuations[13,20,21,40]

$$D(\mathbf{q}, \omega) = \frac{1}{\omega_0(T) + \nu_0 |\mathbf{q} - \mathbf{q}_{CDF}|^2 - i\gamma\omega - (\omega^2/\bar{\omega})}, \quad (2)$$

with the CDF frequency $\omega$ following a parabolic dispersion from $\omega_0$ at $\mathbf{q} = \mathbf{q}_{CDF}$, with coefficient $\nu_0$; $\bar{\omega}$ is the cut-off frequency above which the CDF spectral density decreases more rapidly. The Landau damping parameter $\gamma$ is proportional to the electron density of states that sets a measure of the phase space available for the decay of the fluctuations. It can be shown that in $\mathbf{q} = \mathbf{q}_{CDF}$ the maximum of $Im[D(\mathbf{q}_{CDF}, \omega)]$ is in $\omega = \frac{\omega_0}{\gamma}$, which is thus the energy $\Delta$ directly measured with very high resolution RIXS at the critical wave vector. The minimum frequency $\omega_0$ is also linked to the CDF correlation length $\xi$ by the relation $\omega_0 = \nu_0 \xi^{-2}$, so that it can be independently determined from the width in $q$ of the CDF intensity peak, which is inversely proportional to $\xi$.

At a generic wave vector different from $\mathbf{q}_{CDF}$, the maximum of $Im[D(\mathbf{q}_{CDF}, \omega)]$ is instead reached at $\omega = \gamma^{-1}[\omega_0(T) + \nu_0|\mathbf{q} - \mathbf{q}_{CDF}|^2]$, an energy which is higher than $\Delta$. In particular, when we are far away from

$\mathbf{q}_{CDF}$ the $T$-independent, quadratic term becomes relevant, and this energy is maximum. We have named it $\Omega$.

We have performed a global fit, which simultaneously considers all the YBCO data in both $(H,0)$ and $(H,H)$ directions at the 13 measured temperatures, using Eqs. (1) and (2), with four critical wave-vectors in the first Brillouin zone $\mathbf{q}_{CDF} = (\pm q_{CDF}, 0), (0, \pm q_{CDF})$. The experimental data and the fitting results are compared in Fig. 2b–e. The fitting leads to numerically robust estimates of $\omega_0(T)$, $\nu_0$, $\gamma$ and $\bar{\omega}$ (see Methods). In particular, we find $\omega_0$ to increase from 5 meV at $T_c$ to 20 meV at room temperature (see Fig. 2i), and $\nu_0 = 1.26$ eV (r.l.u.)$^{-2}$ to be close to the value previously found for optimal doping[13,26]. The success of the global fitting with the chosen model entails that the CDF intensity has a major contribution in the quasi-elastic resonant scattering at all $q$ values far from the $\Gamma$ point and causes the almost isotropic increase of its intensity with the temperature, due to the finite energy of the CDF. This means that the whole of the reciprocal space is under the influence of CDF (see Supplementary Fig. 1e).

Given the importance of getting a reliable estimate of $\omega_0(T)$, we have analyzed the data also in a different way. We have isolated the CDF peak close to $\mathbf{q}_{CDF}$ by subtracting, at each $T$, the featureless $(H,H)$ data from the $(H,0)$ ones. This subtraction allows us to remove from the quasi-elastic RIXS intensity the contribution of the elastic scattering due to surface defects, which is independent of the modulus of $q$; at the same time, along the $(H,H)$ direction the CDF contribution is still present though rather flat, so that the shape, i.e. the FWHM, of the CDF

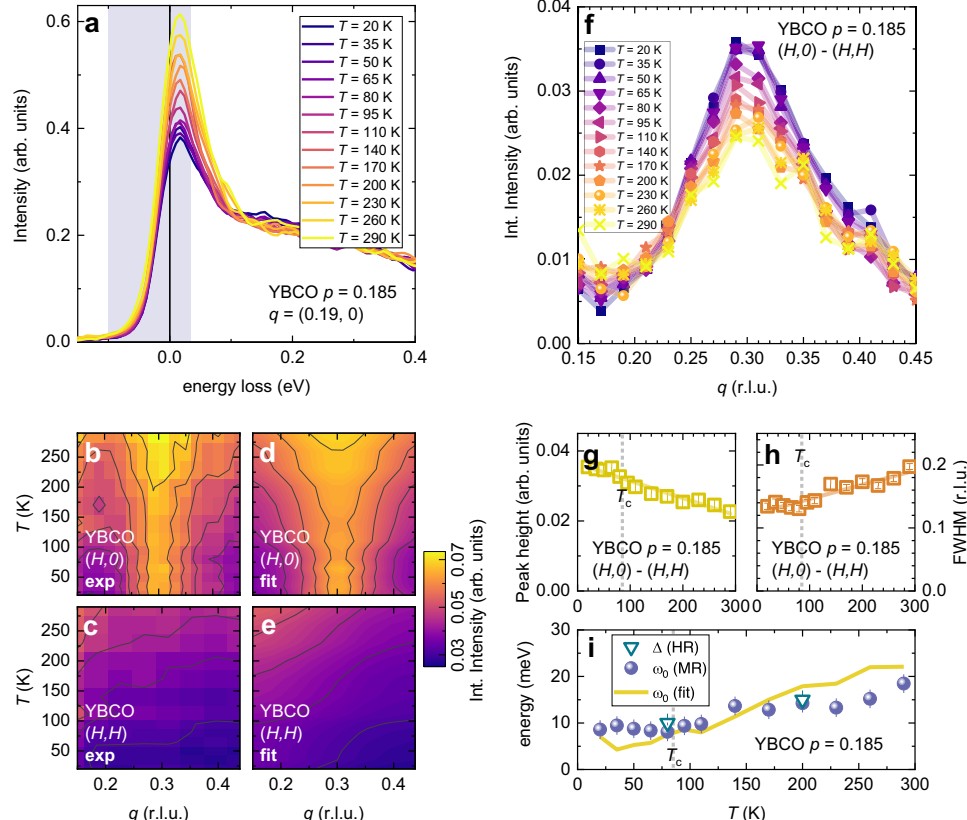

**Fig. 2 | CDF energy from the $T$-dependence of the quasi-elastic spectral weight.**
Medium resolution ($\Delta E = 62$ meV) RIXS spectra have been measured on YBCO
($p \approx 0.185$) at several momenta along both the $(H,0)$ and $(H,H)$ directions, in the
temperature range between $T = 20$ K and $T = 290$ K. **a** RIXS spectra taken at a
representative momentum as a function of the temperature. The vertical band
represents the energy range, $[-0.1, 0.035]$ eV, where the spectral intensity has been
integrated to single out the CDF contribution. **b**, **c** The integrated intensity mea-
sured respectively along the $(H,0)$ and $(H,H)$ directions is shown as a contour plot as
a function of the temperature. **d**, **e** Global fit of the curves presented in panels **b** and
**c**, respectively along the $(H,0)$ and $(H,H)$ direction, achieved by modeling the CDF
peak with the product of the Bose distribution function and of the imaginary part of
the dynamical density fluctuation propagator (Eqs. (1) and (2)). Here, the

experimental data are fitted considering $\omega_0$ varying with temperature in the range
5–20 meV, $\bar{\omega} = 45.55$ meV, $\nu_0 = 1.26$ eV(r.l.u.)$^{-2}$, $\gamma = 1.6$. **f** The CDF peak, given by the
difference $(H,0)$-$(H,H)$, is plotted at several temperatures. **g**, **h** The height and
FWHM of the single Lorentzian profiles used to fit the data in panel **f** are presented
as a function of the temperature. The error bars represent the 95% confidence
interval of the Lorentzian fit. The solid line is a linear fit of the data. **i** The char-
acteristic CDF energy $\Delta$, extracted from the high resolution spectra by the energy
position of the CDF Gaussians, and the frequency $\omega_0$, determined from the medium
resolution spectra by the FWHM of the CDF profiles, are plotted as a function of the
temperature respectively as triangles and circles. Remarkably, the frequency $\omega_0$ at
$q = q_{CDF}$, as determined by the global fit (solid line), is in very good qualitative
agreement with the experiment.

peak is not altered by the subtraction procedure. We have therefore
fitted the resulting $(H,0) - (H,H)$ curves to a Lorentzian peak (Fig. 2f)
and observe that, with increasing $T$, the height of the difference peak
slowly decreases, i.e. the intensity increases less at $q_{CDF}$ than far from it,
and the FWHM increases (Fig. 2g, h). Both $T$ dependences show a mild
slope discontinuity around $T_c$ (see dotted vertical lines in Fig. 2g, h)
which tells us about a possible entwining between CDF and the
superconducting state. Moreover, the extrapolation to zero kelvin of
the high-temperature linear-in-$T$ behavior of the FWHM provides the
estimate of a finite CDF correlation length $\xi \propto FWHM^{-1}$ for $T \rightarrow 0$. This
last occurrence sets a strong difference between CDF and CDW[26] (see
Supplementary Fig. 3c–e), and confirms that a standard criticality
based on the divergence of a spatial correlation length cannot be
contemplated for CDF.

The $\omega_0(T)$ obtained from the FWHM are compared in Fig. 2i with
those determined with the global fit on the same dataset (solid line)
and with the energy $\Delta$ extracted directly from the very-high-resolution
spectra (triangles). Similar energies have been determined for Bi2212
at the same doping level (see Supplementary Fig. 2). The fact that the
values of $\Delta$ and $\omega_0$, determined by different procedures and from
different data sets, coincide within the experimental uncertainties
allows us to take $\gamma \sim 1$ for $T > T_c$; although further investigations might

lead to a more precise estimate of the Landau damping parameter $\gamma$,
we do not expect it to depart significantly from one, if not at low
temperatures, and with superconductivity suppressed by a magnetic
field, as it was also previously found by the fit of the linear-in-$T$ resis-
tivity of YBCO[13].

## Doping dependence of CDF
To determine the dependence of the CDF properties over a sig-
nificantly broad doping range, we performed the RIXS measurements
on a strongly underdoped ($p \sim 0.06$) YBCO sample (see Methods). Even
at such low doping level, the high resolution spectra of Fig. 3a show a
CDF peak, centered at $q = q_{CDF} \sim (0.35,0)$, with an energy
$\Delta(T_{min}, p = 0.06) \sim 25$ meV at 20 K (see Supplementary Fig. 6b, c). This
energy value, so high already close to $T_c$, is therefore significantly
larger than that of the $p \sim 0.19$ samples. Although weaker, the tem-
perature dependence of the CDF intensity and FWHM in the
$p = 0.06$ sample is similar to that of the overdoped ones (see Fig. 3b, c
and Supplementary Fig. 3a, b). We also notice that the softening of the
bond-stretching phonons ($\Delta E_{ph} \sim 4$ meV at 20 K) is still present at
$p = 0.06$, though much weaker than at higher doping (see Supple-
mentary Fig. 6c). This occurrence further supports the generality of
the connection between CDF and bond-stretching phonons.

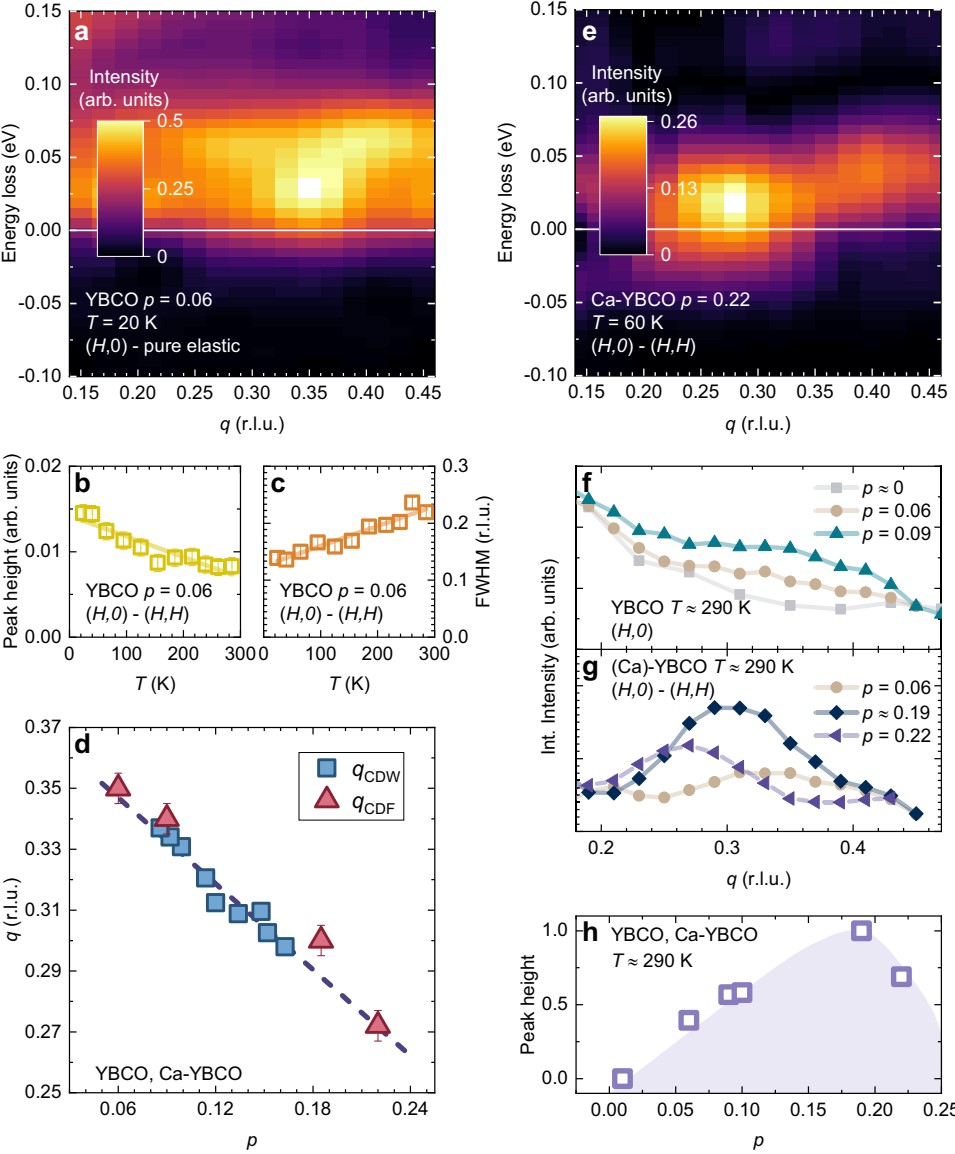

**Fig. 3 | Doping dependence of the charge density fluctuations in YBCO. a** High resolution RIXS map along the $(H,0)$ direction taken at $T = 20\,K$ for a strongly underdoped ($p = 0.06$) YBCO. The map is presented after subtracting the fit of the pure elastic peak from the raw spectra. **b, c** Height and FWHM as a function of the temperature of the Lorentzian profiles used to fit the CDF peak. Here, the peak has been obtained from the difference $(H,0) \cdot (H,H)$ of the intensity of the medium resolution RIXS spectra, integrated within the energy range shown in Fig. 2a. The error bars represent the 95% confidence interval of the Lorentzian fit. The solid line is a linear fit of the data. **d** The $H_{CDF}$ of the CDF peaks in the YBCO and Ca-YBCO samples we have measured is in agreement with the $H_{CDW}$ doping dependence of CDW previously found in YBCO (see squares, taken from ref. 31). The same linear trend is indeed also followed by the samples at $p = 0.06$ and $p = 0.22$, where no

signature of charge order was previously detected. **e** High resolution RIXS map of the difference $(H,0) − (H,H)$ taken at $T = 60\,K$ for an overdoped ($p = 0.22$) Ca-YBCO sample. **f** The integrated intensity at $T \approx 290\,K$ is presented as a function of the momenta along the $(H,0)$ direction for YBCO samples at different level of doping $p$. Curves are presented after a vertical translation, so to align the integrated intensities at the lowest and highest momentum values. **g** The CDF peak, determined by the difference $(H,0) \cdot (H,H)$, is shown at $T \approx 290\,K$ for strongly underdoped ($p = 0.06$) YBCO, for a slightly overdoped ($p = 0.185$) YBCO, and for an overdoped ($p = 0.22$) Ca-YBCO thin film. For clarity, the weak, high-temperature CDF peaks are here presented after smoothing. **h** The height of the CDF peak, normalized to the maximum value measured at $p \approx 0.19$, is plotted vs doping for the YBCO and Ca-YBCO samples. The violet region is a guide to the eye.

For a definitive assessment of the relation of CDF with a quantum critical scenario, we went beyond $p^* \sim 0.19$ by using Ca-doped YBCO thin films with $p = 0.22$ (see Methods). In agreement with previous works[27,31], we find no signatures of CDW. However, we observe a broad and weakly $T$-dependent peak of the quasi-elastic scattering intensity, typical of CDF, centered at $(0.27,0)$. Both this wave vector, and that for $p \sim 0.06$, $(0.35,0)$, fall exactly on the straight line that describes the doping dependence of the CDW wave vector of YBCO[31] (see Fig. 3d). These two values significantly extend the doping range over which the $H_{CDF}$ is known for this cuprate family. The $(H,0) \cdot (H,H)$ RIXS map in Fig. 3e shows that the CDF energy is larger at $p = 0.22$ than at

$p = 0.19$, and the fit of the high resolution spectra leads to $\Delta(T_{min}, p = 0.22) \sim 18\,meV$. Moreover, the CDF scattering intensity is smaller at $p = 0.22$ than at $p = 0.19$. By adding the high temperature data of three other samples (see Fig. 3f, g and Supplementary Fig. 3c–e), at very low ($p \sim 0$) and intermediate doping ($p \sim 0.09$ and 0.10), we can convincingly show in Fig. 3h that the CDF intensity has a maximum at $p = 0.19$. This result is robust and independent of the procedures for the subtraction of the pure elastic contribution.

The results for the $p = 0.06$, $0.19$ and $0.22$ samples suggest that not only the intensity but also the energy associated to the CDF depends on the doping level, calling for a study at intermediate values

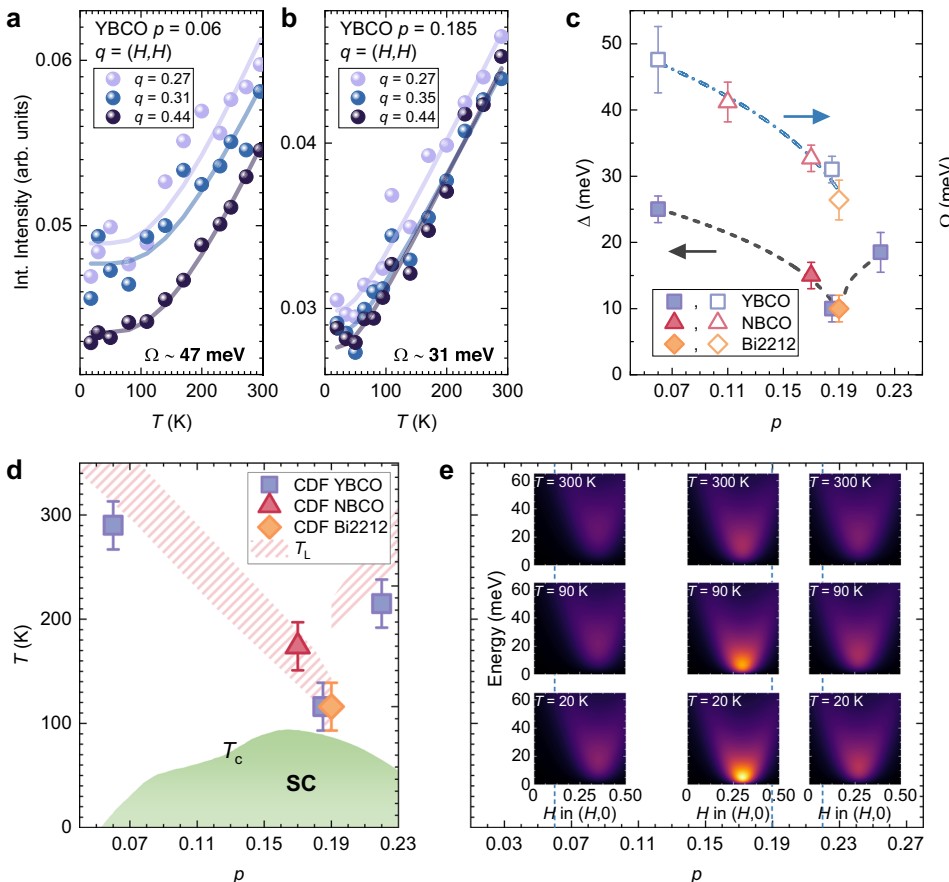

**Fig. 4 | Charge density fluctuations in the cuprate phase diagram. a** The integrated intensity measured on YBCO ($p \approx 0.06$) is presented as a function of the temperature for several momenta along the ($H,H$) direction. For each momentum, the solid line represents the fit of the data assuming a Bose distribution function. **b** Same as previous panel, on YBCO ($p \approx 0.19$). **c** The energies $\Omega$, determined from the Bose fit on spectra measured along the ($H,H$) direction, are plotted together with the energies $\Delta$, directly measured at $q = q_{CDF}$ in the very high resolution spectra. Here and in the next panel we consider the $\Delta$ value measured at the lowest temperature. The two NBCO samples are from Ref. 26. At any doping, $\Omega > \Delta$, as expected when moving away from $q_{CDF}$. As highlighted by the lines, which are

guides to the eye, both energies increase when decreasing the doping, with a minimum at $p = 0.19$. **d** The temperatures corresponding to the energies $\Delta$ are presented as a function of doping $p$ as filled symbols. In the constructed cuprate phase diagram, we also show the temperature $T_L$, where the linear-in-$T$ dependence of the resistance, signature of the strange metal behavior, is lost in YBCO and Bi2212[24,41,42]. **e** In the $p$-$T$ phase diagram, we have depicted the CDF dispersion relation at three temperatures ($T \approx 20$ K, $T \approx 100$ K, $T \approx 300$ K) and doping levels ($p = 0.06$, $p = 0.19$, $p = 0.22$), using the propagator of Eq. (2) and the energy values experimentally determined in this work.

of $p$. However, in those cases the task is more difficult because the CDW peak at zero energy cannot be resolved from the CDF signal at very low energy, being the respective critical wave-vectors almost coincident (see Fig. 3d). Therefore we have studied the CDF far from $q_{CDF}$, where the CDW contribute negligibly to the scattering intensity, i.e., along the ($H,H$) direction and on the tails of the ($H,0$) scan. What we can determine in this way is $\Omega$, the bosonic characteristic energy of the CDF previously mentioned. We expect $\Omega$ to be related to, and larger than, the energy $\Delta$ at $q_{CDF}$. For convenience, we assume that $\Omega$ is constant with respect to temperature, since the $T$-dependent $\omega_0$ term is little relevant in the expression of $\Omega$ for each doping $p$. Consequently, the energy-integral of the quasi-elastic intensity far from $q_{CDF}$ can be simply attributed to a bosonic distribution function of a single characteristic energy $\Omega$ at all temperatures. Even before fitting the experimental data we observe that the $T$ dependence of the intensity is very similar at all $q$ positions along the ($H,H$) and that it strongly depends on the doping level (see Fig. 4a, b). We fitted those curves with a simple function $A + I_0 \left[ 1 + 2 \left( e^{\Omega/k_B T} - 1 \right)^{-1} \right]$ where $I_0$ is the CDF intensity at zero temperature and $A$ accounts for the non-CDF scattering contributions (see Methods and Supplementary Fig. 4). We can thus be confident in using the same method at intermediate doping. Interestingly, $\Omega$ is 15–20 meV larger for $p = 0.06$

than for $p = 0.19$, which is close to what we have also observed at $q_{CDF}$ for $\Delta$ (see Fig. 4c).

The values of $\Delta$, converted into kelvin, are shown in the phase diagram of Fig. 4d for a set of YBCO, NBCO and Bi2212 samples, including those previously used in ref. 26. Both below and above $p^*$ these points line up with the border of the strange metal phase as determined by transport[24,41,42] (shaded regions in Fig. 4d) and define a characteristic wedge with a minimum at $p^*$.

## Discussion

Thanks to an innovative analysis of RIXS data, applied to a large set of measurements (6 doping levels, 3 families of samples, wide temperature range, high and low resolution in energy), we provide here a consistent assessment of the doping and temperature dependences of the CDF intensity and energy in superconducting cuprates. We find that the CDF scattering intensity is strongest in proximity of $p$ ~ 0.19 and at low $T$, while it fades both when increasing the temperature and when moving the doping away from $p^*$. Moreover, the energy $\Delta(T_{min}, p)$, i.e. the minimum of the parabolic relation for the CDF dispersion in the $q$-space, is lowest at $p$ ~ 0.19, while it increases with temperature at all dopings. The results are summarized in Fig. 4e, where we depict the CDF dispersion relation using the propagator of

Eq. (2) calculated with the parameters experimentally determined. We find a good indication that the CDF can be the fluctuations associated to a quantum critical point at $p \sim 0.19$. In fact, the CDF are enhanced exactly at the putative QCP*. Moreover, the CDF energy $\Delta$ coincides, within the experimental errors, to the temperature where the strange metal ends, i.e., where the linear-in-$T$ resistivity is lost. These correspondences point towards a strong role of charge excitations in driving the phase transition, giving rise to the strange metal. However, since recent inelastic neutron scattering and nuclear magnetic resonance results support a quantum criticality at $p^*$ in $La_{2-x}Sr_xCuO_4$ related to spin excitations[43,44], at our present knowledge, we cannot exclude a subdominant spin nature of the quantum fluctuations we have singled out with RIXS. Finally, it is worth reminding that the CDF are strongest at a doping level, slightly above the optimal doping, where the superfluid density[45–47], the superconducting critical current density[46,48] and the upper critical field[49] are the highest.

In a scenario of quantum criticality, one should expect the correlation length of the CDF to diverge at $p^*$ when approaching zero temperature. This is not happening in our data for $p = 0.19$, also because the superconducting order clearly influence the behavior of CDF below $T_c$. We can postulate that, under high magnetic field suppressing superconductivity, the CDF might be drastically different and $\xi$ diverges. However, this scenario is in contradiction with the observation that the linear resistivity of the strange metal extends down to very low $T$ in the absence of superconductivity[3]. In fact, as we attribute the linear resistivity to the quasi-isotropic scattering of carriers by CDF[13], a CDF peak becoming very narrow at low $T$ would not guarantee the needed isotropic scattering mechanism. To resolve this contradiction, we can think either of a frustrated criticality at $p = 0.19$, with $\Delta$ remaining finite as $T \rightarrow 0$, or of $\Delta = \frac{\omega_0}{\gamma} \rightarrow 0$ due to the divergence of the damping factor $\gamma$, with $\omega_0$ staying finite[50]. In the latter case, small domains hosting CDF with finite correlation length $\xi$ would undergo a critical slowing down resulting in an almost persistent glassy state. In other words, the tendency towards a long-range order at the QCP* would be frozen in an incoherent ensemble of CDF droplets. This picture would be consistent with the existence of an anomalous QCP.

Our observations stimulate important questions about the possible interconnection between charge ordering, lattice dynamics, strange metal and superconductivity in cuprates. While the presence of CDF leads to a softening of the high-energy bond-stretching phonons, as we have observed in Fig. 1f, i, it is very plausible a direct coupling of the CDF with the acoustic phonons, whose intensity is robust at the Cu $L_3$ edge[51] and whose energy falls within a similar range as that of CDF[38]. Furthermore, it is noteworthy that the characteristic CDF energy $\Delta(p)$ (see Fig. 4d) is in good quantitative agreement with the $T^*$ line observed in the underdoped regime, up to the doping level $p^*$ where it vanishes[52–54]. However, our experiment cannot unravel the possible cause/effect hierarchy between charge density/quantum fluctuations and the pseudogap[55,56], or the relation of the pseudogap with the quantum criticality[5,52,53]. Conversely, we have shown that at doping levels above $p^*$, i.e., outside the pseudogap region, the charge density fluctuations are still present, and their energy is still in quantitative agreement with the temperature where the strange metal ends, although in the absence of the pseudogap.

## Methods
### Sample growth and characterization
The four $YBa_2Cu_3O_{7-\delta}$ (YBCO) films and the $Y_{0.7}Ca_{0.3}Ba_2Cu_3O_{7-\delta}$ (Ca-YBCO) film, with thickness $t = 50$ nm, have been deposited by pulsed laser deposition on $5 \times 5$ mm$^2$ (001) $SrTiO_3$ substrates. Details of the growth procedure are given in ref. 34. After the deposition, using a post-annealing oxygen pressure of $4.9 \cdot 10^{-5}$, $1.1 \cdot 10^{-4}$, $2.7 \cdot 10^{-4}$, $6.5 \cdot 10^2$ torr (for the YBCO films) and $6.5 \cdot 10^2$ torr (for the Ca-YBCO film), films with a zero resistance critical temperature respectively of 0, 12, 51, 85 K (for the YBCO films) and of 60 K (for the Ca-YBCO film) have been

achieved. The doping levels $p$ have been determined, using a method already successfully used for single crystals[57], by the knowledge of $T_c$ in combination with the $c$-axis length, obtained via X-ray Diffraction: thus, the doping values $p = 0$, 0.06, 0.09, 0.185 (for the four YBCO film) and 0.22 (for the Ca-YBCO film) have been extracted.

The slightly overdoped $Bi_2Sr_2CaCu_2O_{8+\delta}$ (Bi2212) single crystal, with $T_c = 82$ K, has been prepared by growing an optimally doped sample by traveling solvent floating zone method, and by subsequently annealing it in a high-pressure (0.2 MPa) oxygen atmosphere at 500 °C for five days[58].

### RIXS measurements
The RIXS spectra on the slightly overdoped YBCO ($p = 0.185$) and Bi2212 ($p = 0.19$) samples have been collected at the I21 beamline of the Diamond Light Source[59]. The combined (beamline and spectrometer) energy resolution, determined by measuring the width of the non-resonant elastic line on a carbon tape, was 38 meV for the high resolution spectra, measured at two temperatures ($T = 80$ and 200 K) and 65 meV for the medium resolution spectra, measured at twelve different temperatures in the range between 20 and 260 K.

The RIXS spectra on the undoped and underdoped YBCO ($p = 0$, 0.06, 0.09) samples have been collected at the ID32 beamline of the European Synchrotron Radiation Facility (ESRF) in Grenoble using the high-resolution ERIXS spectrometer[60]. The combined (beamline and spectrometer) energy resolution, determined by measuring the width of the non-resonant elastic line from silver paint, was 41 meV for the high resolution spectra (measured at $T = 20$ K, on the $p = 0$ and 0.06 samples) and 62 meV for the medium resolution spectra (measured in the range between 20 and 290 K i) at nine different temperatures for the $p = 0$ and 0.06 samples; ii) at four different temperatures for the $p = 0.09$ sample). Relaxing the energy resolution up to 62 meV has allowed us to perform a thorough temperature dependence. The fully agreement between the results coming from high and medium resolution RIXS spectra, we have presented in the manuscript, demonstrates that the pollution coming from the bond stretching phonons to the CDF dependencies are negligible.

The incident X-rays have been chosen so that their energy is tuned to the maximum of the Cu $L_3$ absorption peak at around 931 eV, and their polarization is linear, perpendicular to the scattering plane ($\sigma$-polarization). They impinge on the sample surface, normal to the YBCO $c$-axis, and are scattered by an angle $2\theta$. The momentum transfers are given in units of the reciprocal lattice vectors $a^* = 2\pi/a$, $b^* = 2\pi/b$, $c^* = 2\pi/c$. We have worked at $2\theta \approx 150°$ in order to get $|q| = 0.91$ Å$^{-1}$, which allowed us to cover the whole first Brillouin zone along the $(H,0)$ direction ($0.5$ r.l.u. $\approx 0.81$ Å$^{-1}$). We have changed the in-plane wave vector component $q_{//}$, fixing $2\theta$ and rotating the samples around $\theta$ only (charge order in cuprates is indeed only weakly $L$-dependent).

The RIXS spectra in the manuscript have been first corrected for self-absorption[61,62] and then normalized to the integral of the inter-orbital $dd$ excitations, in the range $[-3$ eV, $-1$ eV]. The self-absorption correction compensates for the possible reabsorption of scattered photons along their path out of the sample from the scattering point. The probability of self-absorption depends on the absorption coefficients of the incident and scattered photons (which depend on their energy and polarization) and on the scattering geometry. It also depends on the total thickness of the sample. We have implemented the exact form of the correction that takes into account all the parameters: $\theta$ and $\chi$, angles between the normal to the sample surface and the incident and emitted photon propagation direction; $\alpha_{T1,2}$ and $\alpha_{R1,2}$, total and resonant part of absorption coefficient for the incident/scattered photons, with $\alpha_T = \alpha_0 + \alpha_R$ and $\alpha_0$ non-resonant pre-edge absorption coefficient not contributing to the RIXS signal; $d$, sample thickness. The quality of the correction depends on the knowledge of the absorption coefficients, which critically depend on the photon energy and polarization in the proximity of the absorption resonance.

Conversely, the angular dependence is univocally known. The absorption coefficients were evaluated from the XAS spectra measured on every sample during each RIXS experiment. Their relative values are sufficient for the correction in case of thick samples (as single crystals, whose thickness can be approximated to infinite), whereas the absolute value of $\alpha_{T1}$ is needed for thin films, but it can be estimated from tabulated data of $\alpha_0$.

For clarity, we report here the self-absorption correction formulas. Calling $\eta_1(h\nu_2)$ the ideal RIXS spectrum measured with incident photon energy $h\nu_1$ and polarization $\varepsilon_1$, the self-absorption modifies the spectral intensity point by point leading to:

$$I_1(h\nu_2) = \frac{\alpha_{R1}}{\alpha_{T1}} C_1(h\nu_2, \varepsilon_2) D_1(h\nu_2, \varepsilon_2) \eta_1(h\nu_2)$$

with

$$C_1\left(\theta, \chi, \frac{\alpha_{T2}(h\nu_2, \varepsilon_2)}{\alpha_{T1}}\right) = \left(1 + \frac{\alpha_{T2}(h\nu_2, \varepsilon_2) \sin\theta}{\alpha_{T1} \sin\chi}\right)^{-1}$$

$$D_1\left(\theta, \chi, \frac{\alpha_{T2}(h\nu_2, \varepsilon_2)}{\alpha_{T1}}, \alpha_{T1}, d\right) = 1 - \exp\left[-C_1 \cdot \alpha_{T1} \cdot \frac{d}{\sin\theta}\right]$$

We notice that for thick samples, $d \to \infty$ and $D = 1$. Therefore, the spectrum $\eta_1(h\nu_2)$ that ideally would be measured in the absence of self-absorption from a thick sample is derived from the measured one $I_1(h\nu_2)$ as

$$\eta_1(h\nu_2) = \frac{\frac{\alpha_{T1}}{\alpha_{R1}}}{C_1 D_1} I_1(h\nu_2)$$

Thanks to this procedure we can safely compare the shape and intensity of spectra measured at different $\theta$ and $\chi$ angles (i.e., at different **q** points) for a given sample and excitation energy (fixed absorption coefficients, also when the temperature is varied). Spectra of different samples can also be compared, although with some extra precautions due to the uncertainties related to the $\alpha$ coefficients that are not identical in different samples.

We also emphasize here that the self-absorption correction we have applied for both thin films and single crystals does not modify the FWHM and the $H_{CDF}$ of the CDF peak (See Supplementary Fig. 8).

### EI-RXS measurements
Energy Integrated Resonant X-ray scattering (EI-RXS) measurements have been performed at the UE46-PGM1 beamline of BESSY II at Helmholtz Zentrum Berlin[63] on the slightly overdoped YBCO ($p = 0.185$) and Bi2212 ($p = 0.19$) samples. The geometry was identical to that of the RIXS experiments. The samples have been mounted on an ultra-high vacuum diffractometer, in contact with the cold finger of a liquid-Helium-flow cryostat. The incident photons are linearly polarized in the direction perpendicular to the scattering plane (σ-polarization). The scattered photons have been detected using a standard photodiode without discrimination of either polarization and energy, implying that the measured intensities represent an integration over all elastic and inelastic scattering processes. Momentum-space scans have been measured at several temperatures in the range between 10 and 300 K.

### Alignment and fit of the high resolution RIXS spectra
For each sample, we have first determined the zero energy loss line by measuring a resonant spectrum close to $\Gamma = (0,0)$: here, the low energy region is indeed dominated by the purely elastic intensity of the specular, which prevails over any other inelastic contribution, coming from charge order or phonons. The upper error on the position of the

zero can be estimated in the order of 2 meV, as we have shown in detail in the supplementary material of ref. 26. All the spectra measured at higher $q$ values have been aligned among each other, and to the reference spectrum at $\Gamma$, by aligning the maxima of the first derivatives of the quasi-elastic regions.

To extract quantitative information from the spectra, we have fitted those along the $(H,0)$ direction considering the quasi-elastic region as the sum of: i) an energy resolution limited, purely elastic, peak (green Gaussian in Fig. 1c); ii) a low energy peak linked to CDF (red Gaussian in Fig. 1c), whose FWHM is a free fit parameter and whose position is first left free to change, then fixed to its value close to $q_c$, which is more stable (being its intensity overshadowed by that of the specular peak at low $q$ values and by that of the breathing phonons at high $q$ values); iii) a Gaussian associated to the bond-stretching phonons (orange curve in Fig. 1c), whose position and FWHM are free fit parameters; iv) a Gaussian associated to the phonon overtones, with variable FWHM and position set to twice the position of the bond-stretching phonon peak; v) an antisymmetrized Lorentzian function, associated to the paramagnons, as shown in ref. 64.

A similar fit has been done on the spectra measured along the $(H,H)$ direction: for each sample i) the intensity of the purely elastic peak is at any $q$ very close to that along the $(H,0)$ direction (which justifies the difference spectra plotted in Fig. 1a, b) (see Supplementary Fig. 5d); ii) the position of the CDF Gaussian is much higher than that extracted along the $(H,0)$ direction (see Supplementary Fig. 5f), in agreement with theoretical previsions (see Eqs. (1) and (2)), while the intensity of the peak is almost $q$-independent; iii) the bond-stretching phonons do not soften, differently from what observed along the $(H,0)$ direction (see Supplementary Fig. 5h).

Here, it is worth mentioning that the positions in energy of the CDF, $\Delta$, and of the bond-stretching phonons are affected by an uncertainty of about 4 meV. As previously discussed in ref. 26, this is well below the energy resolution of our instrument. Indeed these two peak positions are determined by the difference between the centroid of the pure elastic peak and the centroid either of the CDF peak at $q = q_{CDF}$ or of the bond-stretching phonon peak. This estimation is much more accurate than the instrumental bandwidth: the position of the centroid of a data distribution is well defined and very stable, even when the noise level is relatively high, once the line-shape of the data distribution is well-known.

Our high-resolution spectra could in principle be fitted considering only two peaks in the quasi-elastic region, i.e., the pure elastic peak and the bond-stretching phonon one. However, we have also considered a third peak between these two, which we have assigned to charge density fluctuations. This is mainly for two reasons. First, the fit improves significantly, since we directly account for the presence of the strong intensity at $q_{CDF}$ and finite energy, which is evident from the maps in Figs. 1 and 3. Secondly, it is rather common for Cu $L_3$ RIXS spectra of cuprates, measured with good resolution (35–40 meV BW) to make a three peak fitting for the undoped compounds: one for the elastic component at zero energy, one for the bond-stretching phonons above 50 meV, and one of intermediate energy for other low-energy phonons modes, such as the bond-buckling[65,66]. However, in doped samples, charge order excitations become relevant, and their spectral intensity dominates the region between the elastic and the bond-stretching phonon peak. A three-peak fit (elastic, CDF and bond-stretching) can be performed here, without including an additional, fourth component connected to the bond-buckling phonons. Indeed, their intensity at the Cu $L_3$ edge is much smaller than the other low-energy phonons, as the acoustic and the bond-stretching[51]. We note that, on the contrary, for O $K$ edge RIXS the buckling mode is more intense and cannot be neglected[38,67]. Moreover, in the range of $q$ where the CDF is the strongest and its energy $\Delta$ is determined, i.e. at $q$ close to $q_{CDF} \approx 0.25$-$0.35$ r.l.u.[51], their intensity, being proportional to $\cos^2(\pi q)$[51], is even weaker. Therefore, we decided not to include the buckling

phonon contribution in our fitting, in order to restrict the number of free parameters and obtain more robust results. For completeness, we have anyway made a test with four peaks, adding a fourth Gaussian component, centered at the energy where buckling phonons have been previously measured in YBCO and Bi2212 by other techniques. We found that the intensity of this additional phonon mode is low enough to leave unaltered the position of the CDF contribution, i.e., the CDF energy $\Delta$ (see Supplementary Fig. 9). Of course, a softening of the bond-buckling phonon branch due CDF is expected, as it has been already measured in cuprates at the O $K$ edge[38,67]. However, the intensity we measure at the Cu $L_3$ edge is too low to study this phonon anomaly and to attribute to this phonon anomaly any change in the CDF parameters as extracted by the three-peak fit.

### Global fit of the medium energy resolution RIXS spectra

Medium energy resolution RIXS spectra have been integrated over the [−100, 35] meV energy range to extract quantitative information on the intensity evolution of the CDF peak with momentum $q$ and temperature $T$, irrespective of the details of its spectral line shape, as shown in Fig. 2a. The $I(q,T)$ curves were all fit simultaneously to extract robust estimates of the parameters appearing in Eqs. (1) and (2), as shown for slightly overdoped YBCO (see Fig. 2b–e and Supplementary Fig. 1a, b) and Bi2212 (see Supplementary Fig. 2a, b).

### Bose fit of the medium energy resolution RIXS spectra

The Bose function $A + I_0 \left[ 1 + 2 \left( e^{\Omega/k_B T} - 1 \right)^{-1} \right]$ used in the main text depends, among the other, on the values of the variables $A$ and $I_0$. Their sum represents the total quasi-elastic intensity in the zero temperature limit: $I_0$ is the CDF intensity, which far from $\mathbf{q}_{CDF}$ we consider $q$-independent, while $A$ accounts for the non-CDF contributions, which is strongly $q$-dependent, and minimum at high $q$ values (consequently to the $q$-dependence of the specular elastic peak, see Supplementary Fig. 6a). For each sample, we have determined the $A$ and $I_0$ values from the high-resolution spectra, given the relative weight of the purely elastic Gaussian and of the CDF Gaussian in the [−100, 35] meV energy range. The doping dependence of the $\Omega$ values, i.e. CDF energies along the $(H,H)$ direction estimated from the fit, is however very robust, and the solidity of the $\Omega$ values goes beyond any possible uncertainties in the correct determination of $A$ and $I_0$.

In Fig. 4a, b the difference between the data taken on samples at various dopings is very clear. In particular, the slopes of the quasi-linear trend of the integrated intensity versus temperature are very different (see also Supplementary Fig. 4a). In a Bose scenario, this means that the energies $\Omega$ are different. More precisely, we can estimate the ratio between the $\Omega$ values at different dopings, which is a good indicator of the situation. This approach is based on the fact that the Bose function can be put in a universal form by plotting it in terms of the reduced temperature $T/\Omega$. Thus, different curves will collapse onto the universal function if they are plotted as a function of $T/\Omega$. Because the scaling factors are different in the two cases, the definition of reduced temperature, thence the energy $\Omega$, is different. This is very clear from Supplementary Fig. 4b, where we have plotted as a function of $T/\Omega$ the two curves of the strongly underdoped and of slightly overdoped samples at $q = 0.44$ r.l.u. Here, the ratio is 1.5, which is in agreement with the $\Omega_{p=0.06}/\Omega_{p=0.19}$ ratio (47/31), resulting from the Bose fit presented in the main text. This ratio of the two scaling factors does not depend on the choice of one of the two values of $\Omega$, and is $T$-independent because of the universality of the Bose plot.

Focusing instead on the absolute values of $\Omega$, the values determined from the fit are in fairly good agreement with the values we have directly measured in the high-resolution spectra. Indeed, the position of the CDF Gaussian along the $(H,H)$ direction is, depending on the $q$ value, in the 35–38 meV energy range (see Supplementary Fig. 5f): this value is only ∼15% higher than that we have found using the Bose fit.

## Data availability

All data shown in the main text and in the supplementary information are available at the Zenodo repository[68], under accession code https://doi.org/10.5281/zenodo.8430044.

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

## Acknowledgements

The authors acknowledge Zhi-Xun Shen, Tom Devereaux, Wei-Sheng Lee, Jan Zaanen, Ulf Gran and Alexander Krikun for insightful discussions, Roberto Fumagalli for helping with EI-RXS measurements, and Alexei Kalaboukhov for the support with the PLD used for the deposition of the Ca-YBCO thin films. This work was performed in part at Myfab Chalmers. The RIXS experimental data were partly collected at the beam line ID32 of the European Synchrotron (ESRF) in Grenoble (France) using the ERIXS spectrometer designed jointly by the ESRF and Politecnico di Milano. We acknowledge Diamond Light Source for providing the beam time at the I21-RIXS beamline under proposal MM23880. We thankfully acknowledge HZB for the allocation of synchrotron radiation beam time, at the beamline UE46-PGM1 of BESSY II where the EI-RXS measurements have been carried out. This work is supported by the project PRIN2017 "Quantum-2D" ID 2017Z8TS5B of the Ministry for University and Research (MIUR) of Italy (L.M., M.M.S., S.C., M.G., G.G.), by the Swedish Research Council (VR) under the projects 2018-04658, 2022-04334 (F.L.), 2020-04945 (R.A.) and 2020-05184 (T.B.), by the European Union's Horizon 2020 research and innovation programme (grant agreement No 730872). S.C. and M. G. also acknowledge financial support from the University of Rome Sapienza, through the projects Ateneo 2020 (Grant No. RM120172A8CC7CC7), Ateneo 2021 (Grant No. RM12117A4A7FD11B), Ateneo 202 (Grant No. RM12218162CF9D05). M.G. acknowledges financial support from PNRR MUR project PE0000023-NQSTI.

## Author contributions

R.A., L.B. and G.G. conceived and designed the experiments with suggestions from F.L., C.D.C. and M.G. L.M., R.A., M.M.S., G.G., L.B., N.B.B., A.N., K.J.Z., M.G.-F., Q.L. and Y.Y.P. performed the RIXS measurements. E.S., R.A., M.M.S., L.M. and G.G. performed the EI-RXS measurements. R.A., T.B. and F.L. grew and characterized the YBCO films; Q.G. and X.Z. grew and characterized the Bi2212 single crystal. R.A., L.B., M.M.S., P.C. and G.G. analyzed the RIXS experimental data; R.A., L.B., G.G, M.M.S., S.C., C.D.C., M.G., F.L., T.B. and L.M. discussed and interpreted the results. R.A. and G.G. wrote the manuscript with the input from M.G., C.D.C., F.L., L.B., M.M.S., S.C. and contributions from all authors.

## Funding

## Competing interests

The authors declare no competing interests.
