## [Peer Review File · Nature Communications]

REVIEWER COMMENTS

Reviewer #1 (Remarks to the Author):

The main claim of this paper is that charge fluctuations have a quantum critical character and are responsible for the strange metal behavior in YBa₂Cu₃O_{7-d} (data limited to a single doping level in Bi₂212 is presented too). From this, the authors speculate that these fluctuations may even be responsible for the superconducting pairing and that these considerations likely apply to all cuprates.

There is hardly a more important question in the field and there is no doubt that the authors present very interesting data.

Essentially, the argument is that the energy of charge fluctuations decreases with increasing doping while the intensity of these fluctuations increases. My main problem with this is the absence of data for doping levels beyond $p=0.19$ that is supposed to be the quantum critical point. Therefore, it is unknown whether the characteristic energy and the intensity reach their minimum/maximum at $p=0.19$ or whether they still evolve with doping beyond $p=0.19$. Both cases are problematic with respect to quantum criticality: in the first case, it is unclear whether a drop of the energy by a factor 2 and an increase of the intensity by a factor of 2 or 3 (not sure how to interpret Fig. 3e) are enough to establish quantum criticality. In the second case, there cannot be a QCP at $p=0.19$.

Adding to this concern is that there seems to be evidence of charge density waves and/or charge density fluctuations beyond $p=0.19$ in various cuprates. Furthermore, as acknowledged by the authors themselves, they are not probing the standard signatures of quantum criticality with an ordered phase disappearing at the QCP. So, all in all, I find it hard to be convinced that the presented data provide solid evidence of any sort of quantum criticality (although I am not saying this is excluded).

If the authors can rule out these objections or can add higher doping data, I would certainly be willing to consider a revised version of their manuscript.

I find the link between charge fluctuations and strange metal behavior more convincing but as far as I understand this has been discussed by the authors in previous publications.

Reviewer #2 (Remarks to the Author):

Arapia et al present an extensive study of RIXS measurements in overdoped YBCO and BSCCO for the purpose of extracting evidence of charge density fluctuations (CDF).

First I want to remark that this is an important endeavor. Charge density fluctuations are plausible feature of the cuprate phase diagram and may indeed be connected to a quantum critical point in the cuprate phase diagram. If this is the case, it is of fundamental importance to the physics of the cuprates, and of general interest to the researchers in superconductivity and correlated materials, to understand how charge density fluctuations evolve with doping and the degree to which they are connected to a quantum critical point. Moreover, observing charge density fluctuations is challenging requiring technically difficult experiments and novel approaches to data analysis. Quantifying the properties of charge density fluctuations as a function of temperature and doping is more challenging than observing evidence for charge density fluctuations, and is a central aim of this manuscript.

From these efforts, the authors report a few important observations:

1. The energy of charge density fluctuations decreases with increasing doping, consistent with what one would expect approaching a quantum critical point.
2. CDF scattering is observed throughout reciprocal space and can be fit as a function of temperature and q , to functional form that is given by quantum fluctuations.

Despite the potential interest in these findings and experimental challenges, however, I think the present study has several shortcomings that cast doubt on these conclusions. Because of these shortcomings, detailed below, I do not think the paper provides the clarity required for publication in an upper tier journal such as Nature Communications.

My concerns are the following:

1. One of their principle findings is that CDF are present at the CDW wavevector, but also throughout reciprocal space. Specifically, they conclude that the “The success of the global fitting with the

chosen model entails that the CDF contribution dominates the quasi-elastic resonant scattering at all q values far from the Γ point and causes the almost isotropic increase of its intensity with the temperature, due to the finite energy of the CDF. This means that the whole reciprocal space is influenced by the CDF.”

In other words, the authors claim that all other contributions to diffuse scattering are small relative to CDF and can be ignored. However, despite the seeming success of their fitting, it is not clear to me that this statement is correct.

However, there must be other contributions to diffuse scattering in these systems whose origins are not associated with charge density order and the fluctuations of charge density wave order. Namely, there is both thermal diffuse scattering (TDS) from lattice vibrations and diffuse scattering from structural disorder in the materials, including but not limited to ortho oxygen ordering in YBCO, should occur and in principle contribute to the measured signal. Moreover, thermal diffuse scattering has a temperature dependence given by the thermal population of low energy phonon modes. This is often expressed to have an intensity dependent on ω/kBT , and may be qualitatively similar to the Bose occupation factor T dependence shown in the paper. As such, it is not evident that a more conventional explanation cannot explain at least some of the temperature dependent scattering.

Associated with this concern about other sources of scattering, the authors show in fig. 3d data on an $p = 0$ sample. However, temperature dependent data does not appear to be shown. This would presumably provide some insight, as one would assume that charge density fluctuations are not present at that doping. Is temperature dependence seen for the quasi-elastic scattering for the $p = 0$ sample?

2. In light of the proposed description of all of the quasi elastic scattering as originating from charge density fluctuations, I find some of the subsequent analysis confusing. The authors argue that the CDF can be fit to a functional form given by equations 1 and 2, with the whole data set along both (H,H) and (H,0) fit with this form + some T independent but q dependent background. The functional form of equations 1 and 2 is neither a Gaussian in energy or a Lorentzian in q . Despite this, the authors have fit the CDF peak to a Gaussian in energy and the q dependence to a Lorentzian, neither of which is consistent with eq. 1 and 2. Moreover these fits are done after subtracting the (H,H) scans from the (H,0) scans. This would be appropriate if the (H, 0) scans is on top of a background that is isotropic and equal to that from the (H,H) scan. However, the authors claim that the HH scans contain CDF and are thus not completely “background”. These inconsistencies provide for a somewhat confusing narrative. More significantly, one worries that these various subtractions and fits are somehow yielding incorrect results.

3. The main conclusions are only achieved after extensive multi-peak fitting of the energy and Q dependence to the high resolution data. However, the key “peak” of interest, the CDF peak, is the least resolved in raw data, being sandwiched in energy between elastic scattering and phonon modes, and also overlapping in energy with a paramagnon peak. As with this sort of fitting, the error bars on fitted peak parameters often do not provide a good representation of the certainty of the results. That is because there are often a range of parameter space that has similar reduced χ^2 values. Accordingly, a different choice of fitting procedure, such as using eqn. 1 instead of a Gaussian, or Voigt functions instead of Gaussians, or adding a bond-buckling phonon into the mix, would likely alter important parameters extracted from these fits and their doping dependence. While some of this may be inconsequential for the main story of the paper, it is not unreasonable to think that variation in fitting procedure could affect a key parameter, such as Δ , the energy of the CDF peak.

4. The authors state that they correct all of the data for self-absorption corrections. However, the self-absorption corrections is different for a thin film vs a bulk single crystal. Moreover, these corrections can be substantial depending on the geometry, possibly introducing a systematic error that dominates the q dependence of the scattering. Similar to concern #3 above, systematic errors in this procedure can impact quantities such as the values of the FWHM of a peak or the peak position. Can the authors provide more details of the absorption correction? Was this done considering thin films in the case of YBCO and bulk crystals in the case of Bi2212.

Reviewer #3 (Remarks to the Author):

This is an interesting piece of work. The authors claim that the charge density fluctuations (CDF) observed from the resonant x-ray scattering possess the right properties to be associated to a quantum phase transition. The authors have studied the CDF in two families of cuprate superconductors over a wide range of doping p , up to $p^* \approx 0.19$, corresponding to the putative quantum critical point (QCP).

The possible relation to quantum criticality and strange normal and superconducting state properties of hole doped high- T_c cuprates are a matter of intense debate. Since the early days of the proposal made by Loram, Tallon, and Castellani, there have been numerous studies. Most of these studies are concerned with the pseudogap (PG) observed in the hole doped cuprates. The possible interrelations among pseudogap, charge/spin density fluctuations, Gaussian superconducting fluctuations (Nernst effect) and superconductivity are not clear yet. Against this backdrop, the present work is important. The experimental results and the analysis seems sound. I am in favor of publication of this paper. But there are few issues that require further clarification before this manuscript (m/s) can be accepted.

1. There are evidences that the CDF and PG are not related directly (Phys. Rev. Lett. 94, 237002 (2005); Physica C 387, 365 – 372 (2003); Supercond. Sci. Technol. 21 105017 (2008); Supercond. Sci. Technol. 21 125020 (2008)). At the same time, the PG vanishes at $p \sim 0.19$ (phys. stat. sol. (b) 215, 531 (1999); Physica C 349, 53 (2001); Phys. Rev. B 71, 054502 (2005); Scientific Reports volume 9, Article number: 14856 (2019)). This is the same hole content reported in this paper. It cannot be a coincidence. The authors need to address this issue.

2. Charge ordering leads to the $1/8$ th anomaly which degrades the superconducting transition temperature. The hole content dependent PG energy shows featureless behavior close to the $1/8$ th doping (Phys. Rev. B 71, 054502 (2005); Supercond. Sci. Technol. 21 105017 (2008); Supercond. Sci. Technol. 21 125020 (2008)). A CDF analysis close to $p \sim 0.125$ will be informative. The authors should look at this matter.

3. In Fig. 4d, the $\Delta(p)$ line looks very similar to the PG(p) line (Phys. Rev. B 71, 054502 (2005); Physica C 387, 365 – 372 (2003); Physica C 476, 10 (2016)). This resemblance needs addressing. It should be noted that the PG(p) line may not be associated with any symmetry breaking (distinct phase) in the thermodynamic sense.

Response to referees

We thank the referees for their overall positive and very constructive reports. We are convinced that we can overcome all criticism and that the manuscript has been significantly improved thanks to the input by the reviewers. In what follows we reply point by point to all referees' comments and suggestions, and we indicate the modifications made to the text (in the revised version, the major modifications have been highlighted in blue for clarity):

Reviewer #1:

The main claim of this paper is that charge fluctuations have a quantum critical character and are responsible for the strange metal behavior in $\text{YBa}_2\text{Cu}_3\text{O}_{7-d}$ (data limited to a single doping level in Bi_2Te_2 is presented too). From this, the authors speculate that these fluctuations may even be responsible for the superconducting pairing and that these considerations likely apply to all cuprates.

There is hardly a more important question in the field and there is no doubt that the authors present very interesting data.

We thank the referee for appreciating our RIXS data and for understanding the relevance and the potential impact of our research.

Essentially, the argument is that the energy of charge fluctuations decreases with increasing doping while the intensity of these fluctuations increases. My main problem with this is the absence of data for doping levels beyond $p=0.19$ that is supposed to be the quantum critical point. Therefore, it is unknown whether the characteristic energy and the intensity reach their minimum/maximum at $p=0.19$ or whether they still evolve with doping beyond $p=0.19$. Both cases are problematic with respect to quantum criticality: in the first case, it is unclear whether a drop of the energy by a factor 2 and an increase of the intensity by a factor of 2 or 3 (not sure how to interpret Fig. 3e) are enough to establish quantum criticality. In the second case, there cannot be a QCP at $p=0.19$.

We completely agree with the referee about the crucial importance of data at doping levels above $p^*=0.19$ in order to establish the presence of a quantum criticality at p^* . The lack of these data in the first version of the manuscript was simply due to the limited accessibility to RIXS facilities, which didn't allow us to allocate any beamtime for this RIXS experiment (which we proposed, without success, one year ago) before the first submission. We have filled this gap in the new version of the manuscript. Indeed, we have measured high resolution RIXS spectra of strongly overdoped ($p=0.22$) YBCO, in a very broad range of momentum, and at two different temperatures (T_c and room temperature). The outcome of the measurements has been summarized in Figures 3 (panels d-e-g-h) and 4 (panels c-d-e), which have been deeply modified in the revised version. The main results (described in details in the "Doping dependence of the CDF" subsection of the revised manuscript) are the following: at $p=0.22$, long-ranged, quasi-static, charge density waves are missing, but dynamic charge density fluctuations still survive: their energy Δ is of the order of 20 meV, i.e., almost a factor 2 higher than at $p=p^*=0.19$, while their intensity is very weak, i.e., about 30% lower than at $p=p^*=0.19$. These additional measurements confirm therefore that $p=p^*$ is a special doping level, where the energy is minimum, and the intensity is maximum: this occurrence significantly strengthens therefore the conclusions of our manuscript.

The new data at $p=0.22$, together with those we previously took at $p=0.06$, also allow us to make another observation. The CDF wave vector at $p=0.22$ ($q=0.27$ r.l.u.), and that at $p \approx 0.06$ ($q=0.35$ r.l.u.) fall exactly on the straight line that describes the doping dependence of the CDW wave

vector of YBCO [Phys. Rev. B. 90, 054513 (2014)]. These two values significantly extend the doping range over which the charge order wavevector is known for this cuprate family. We have therefore added this discussion in the “Doping dependence of CDF” subsection of the revised manuscript, and made a new panel, 3d, which underlines this outcome.

Finally, pushed by the referee comment about the difficulty to determine the doping dependence of the CDF peak intensity from Figure 3 of the previous manuscript, we have added a new panel in Fig 3 of the revised manuscript, panel 3h, where the height of the CDF peak (determined as the difference between the H0 and the HH scan), normalized to the height of the CDF peak at p^* , is shown as a function of the doping in the range $0 < p < 0.22$. This figure, together with Fig. 4c, where the doping dependence of the CDF energy is shown, summarizes the main findings of our measurements: this should favor the readability of the paper and highlight the peculiarity of the doping level $p=p^*$.

Adding to this concern is that there seems to be evidence of charge density waves and/or charge density fluctuations beyond $p=0.19$ in various cuprates. Furthermore, as acknowledged by the authors themselves, they are not probing the standard signatures of quantum criticality with an ordered phase disappearing at the QCP. So, all in all, I find it hard to be convinced that the presented data provide solid evidence of any sort of quantum criticality (although I am not saying this is excluded).

The additional measurements we have included in the revised manuscript confirm that charge density fluctuations also occur above p^* , although they are weaker and their energy is higher than at p^* , where the energy and the intensity respectively have their minimum and maximum. As mentioned by the referee, a broad-in-q peak, signature of charge fluctuations, has been observed above p^* also in other cuprate compounds as Bi2212 [Phys. Rev. B 106, 155109 (2022)] or LSCO [npj Quantum Materials 6, 31 (2021)]. However, a systematic study of the doping and temperature dependence of the CDF energy and intensity, on which our paper is based, has never been carried out, so the crucial role of the doping level $p=p^*$ in relation to charge fluctuations could not have assessed.

The referee also asks about the possible presence of a narrow-in-q peak, signature of charge density waves, above p^* . Our new measurements at $p=0.22$ tell us very clearly that CDW are absent above p^* in the YBCO family. The CDW phenomenology in different cuprate families confirms to be very heterogeneous, as already observed during the last two decades: indeed, a narrow-in-q charge order peak, in connection to a Van Hove singularity in the overdoped region has been observed in Bi2201 [Nat. Mater. 17, 67 (2018)], and a second CDW dome has been observed at low temperatures in the overdoped region of LSCO [npj Quantum Materials 6, 31 (2021)], even though in the latter case the results are still controversial [npj Quantum Materials 8, 7 (2023)]. However, the discussion about the CDW goes beyond the focus of our manuscript, so we didn't add on this in the revised version.

Finally, as mentioned by the referee we acknowledge in the manuscript that the quantum criticality we are probing is anomalous. This is because the energy we measure at $p=p^*$ is minimum but still finite. On the other hands if at $p=p^*$ there is a QCP, Δ should vanish at $T=0$ and have a minimum as a function of doping at finite T as our new measurements confirm. In standard quantum criticality Δ vanishes because the correlation length goes to infinity. Here the measured correlation length extrapolates to a finite value at $T=0$ (see Fig. 2h). In the discussion section, we argue this anomaly could be due to the damping factor γ , possibly diverging at p^* when superconductivity is suppressed and therefore bringing the energy Δ to zero. Unfortunately, there is no RIXS experimental setup equipped with magnetic fields able to suppress the superconducting state at p^* , so our data cannot confirm this theoretical scenario. As a consequence, to cover a broader range of possibilities, we have mentioned in the discussion section of the revised version an alternative scenario of anomalous quantum

criticality, i.e., a frustrated quantum criticality, where the damping factor is not diverging and the energy stays finite even at $T \rightarrow 0$ and $p \rightarrow p^*$, in absence of superconductivity. Moreover, in the revised version we have discussed more thoroughly about γ when presenting the theoretical model (page 6), given the importance of γ in the final discussion section.

If the authors can rule out these objections or can add higher doping data, I would certainly be willing to consider a revised version of their manuscript.

We believe the new data we have added in the revised version at $p=0.22$, and the observed doping and temperature dependencies of the CDF energy and intensity, have strengthened our paper and strongly supported our conclusions. Thence, we hope this revised version can satisfy the referee, ruling out his objections.

I find the link between charge fluctuations and strange metal behavior more convincing but as far as I understand this has been discussed by the authors in previous publications.

Before the current manuscript, the link between charge fluctuations and strange metal was only theoretically predicted, but never experimentally proven. Indeed, in our first publication on CDF [Science 65, 906 (2019)] we have only shown the presence of these very short ranged, dynamical modulations in a broad region of the T - p cuprate phase diagram, including the strange metal region. After that, it has been theoretically predicted, and numerically confirmed, by a subset of the co-authors of our manuscript, that - being broad-in- q - CDF could provide the isotropic scattering rate on which the linear-in- T resistance in the strange metal is based [Commun Phys 4, 7 (2021)]. Here, instead, we show that that the energy of the CDF, converted in temperature, follows vs p the same lines, both below and above p^* , where the strange metal ends in the cuprate phase diagram. Consequently, only the data presented in the current manuscript can set a definitive link between charge fluctuations and strange metal behavior.

In addition to that, the doping and temperature dependencies of the CDF energy and intensity allowed us to go one step further, concluding that CDF look compatible with the quantum fluctuations emerging from an (anomalous) QCP at $p=p^*$, which is therefore connected - through the abovementioned link - to the strange metal regime.

Reviewer #2:

Arpaia et al present an extensive study of RIXS measurements in overdoped YBCO and BSCCO for the purpose of extracting evidence of charge density fluctuations (CDF). First I want to remark that this is an important endeavor. Charge density fluctuations are plausible feature of the cuprate phase diagram and may indeed be connected to a quantum critical point in the cuprate phase diagram. If this is the case, it is of fundamental importance to the physics of the cuprates, and of general interest to the researchers in superconductivity and correlated materials, to understand how charge density fluctuations evolve with doping and the degree to which they are connected to a quantum critical point. Moreover, observing charge density fluctuations is challenging requiring technically difficult experiments and novel approaches to data analysis. Quantifying the properties of charge density fluctuations as a function of temperature and doping is more challenging than observing evidence for charge density fluctuations, and is a central aim of this manuscript. [...]

Despite the potential interest in these findings and experimental challenges, however, I think the present study has several shortcomings that cast doubt on these conclusions. Because of these shortcomings, detailed below, I do not think the paper provides the clarity required for publication in an upper tier journal such as Nature Communications.

We thank the referee for understanding the relevance and the potential impact of our research and the challenges behind the data we have presented. We are confident we have replied to all

his/her concerns, and therefore hope the revised version clarifies his/her doubts and overcome his/her criticism.

1. One of their principle findings is that CDF are present at the CDW wavevector, but also throughout reciprocal space. Specifically, they conclude that the “The success of the global fitting with the chosen model entails that the CDF contribution dominates the quasi-elastic resonant scattering at all q values far from the Γ point and causes the almost isotropic increase of its intensity with the temperature, due to the finite energy of the CDF. This means that the whole reciprocal space is influenced by the CDF.”

In other words, the authors claim that all other contributions to diffuse scattering are small relative to CDF and can be ignored. However, despite the seeming success of their fitting, it is not clear to me that this statement is correct.

However, there must be other contributions to diffuse scattering in these systems whose origins are not associated with charge density order and the fluctuations of charge density wave order. Namely, there is both thermal diffuse scattering (TDS) from lattice vibrations and diffuse scattering from structural disorder in the materials, including but not limited to ortho oxygen ordering in YBCO, should occur and in principle contribute to the measured signal. Moreover, thermal diffuse scattering has a temperature dependence given by the thermal population of low energy phonon modes. This is often expressed to have an intensity dependent on ω/kBT , and may be qualitatively similar to the Bose occupation factor T dependence shown in the paper. As such, it is not evident that a more conventional explanation cannot explain at least some of the temperature dependent scattering.

We thank the referee for giving us the opportunity to elaborate more on this important point. When analyzing the spectra, we need to take into account that in RIXS the diffuse scattering from structural disorder is in general smaller than in other non-elastic scattering techniques. In particular in our case we can exclude the contribution of ortho oxygen ordering to the diffuse scattering: YBCO, Ca-YBCO, NBCO are in thin film form, where the chain order, differently than in single crystals, is not preserved; Bi2212 is in single crystal form, but chains are absent in its crystal structure. Of course, a diffuse scattering contribution has to be present, and it is not irrelevant: it is at the origin of the pure elastic peak which we have explicitly singled out in the fit shown in Fig. 1, whose intensity is maximum at Γ . But this peak, as shown in Fig. 1d and 1g, is T -independent, so it cannot modify neither the temperature dependence of the CDF energy as determined by the global fit, and shown by the yellow curve in Fig. 2i, nor Ω , i.e. the energy value of CDF far from q_{CDF} , where we have explicitly considered a T -independent term, A , taking into consideration non-CDF scattering contributions.

Some problems to the correct determination of Ω from the “Bose fit” could in principle arise from the thermal diffuse scattering from phonons, as correctly pointed out by the referee. This contribution should be relevant for low-energy (few meV) phonons. However, we can exclude that this is detrimental for our analysis of Ω : first, this contribution, hence the RIXS cross sections of these phonons, seems small, otherwise we should have observed a pronounced T -dependence in the pure elastic peaks determined in Fig. 1; second, this contribution, if significant, should be almost doping independent, while we observe a strong doping dependence for Ω . In the worst hypothesis, a significant thermal diffuse scattering term should just imply a rigid shift of the Ω values at all p levels: this would not be a problem for us, since we are not interested in the absolute values of Ω but only in their p -dependence (to further corroborate the measured p dependence of Δ , the energy at q_{CDF} , which is the parameter having greater relevance.).

Finally, we also recall here that the main reason we believe our global fit of the H0 and HH data via equations 1 and 2 is successful, despite the evident – even brutal – approximations we have made, is provided by the determination of the CDF energy, which is in very good quantitative agreement with the energy values directly determined from the high resolution RIXS spectra at q_{CDF} , both at low and at high temperatures.

Associated with this concern about other sources of scattering, the authors show in fig. 3d data on an $p = 0$ sample. However, temperature dependent data does not appear to be shown. This would presumably provide some insight, as one would assume that charge density fluctuations are not present at that doping. Is temperature dependence seen for the quasi-elastic scattering for the $p = 0$ sample?

In principle, we agree with the referee that a completely undoped sample can be a good benchmark to prove the presence and quantify the thermal diffuse scattering contribution in the RIXS spectra due to phonons. However, removing the whole hole doping from superconducting cuprates is very tough, which is the reason why – already in the previous version of our manuscript - we associate to the insulating sample the doping level $p \approx 0$. This detail is of particular relevance since STM studies proved clear evidence of a precursor, possibly commensurate, charge order state already in the insulating state of HTS cuprates, as soon as the doping p is not zero [Nat. Phys. 12, 1047 (2016); Nat. Mater. 18, 103 (2019)].

The quasi-elastic intensity of our $p \approx 0$ YBCO sample still exhibits a mild temperature dependence (with the intensity growing when T increases). Up to now, we cannot say if this signal increase is due to thermal diffuse scattering from low-energy phonons or if this represents the first confirmation, from a technique different than STM, of a precursor charge order in the insulating state of HTS cuprate. These measurements are currently object of investigation from our side, but the results are however beyond the scope of the current manuscript. Indeed, as also previously stated, even if this T -dependent contribution would originate from thermal diffuse scattering of low-energy phonons, that would not be an issue for us and it would not modify the conclusions of our manuscript. This putative thermal diffuse scattering contribution would be almost doping independent and would not modify the doping dependence we have determined for Ω , but only the absolute values (which are not relevant, as explained in the previous reply to the same referee).

2. In light of the proposed description of all of the quasi elastic scattering as originating from charge density fluctuations, I find some of the subsequent analysis confusing. The authors argue that the CDF can be fit to a functional form given by equations 1 and 2, with the whole data set along both (H,H) and (H,0) fit with this form + some T independent but q dependent background. The functional form of equations 1 and 2 is neither a Gaussian in energy or a Lorentzian in q. Despite this, the authors have fit the CDF peak to a Gaussian in energy and the q dependence to a Lorentzian, neither of which is consistent with eq. 1 and 2.

It is actually true that the peaks given by Eqs 1 and 2 are not Lorentzian nor gaussian. However, both the fitting procedures we have followed are correct:

i) in Fig. 1, we have to fit the CDF peaks to a gaussian in energy, since the peaks as given by Eqs 1 and 2 have to be convoluted with the combined (beamline and spectrometer) experimental energy resolution, which is quite large and Gaussian. This is why the line shape has to be Gaussian in energy.

ii) in Fig. 2, we have used a Lorentzian fit for the CDF peak in q since the Eqs 1 and 2, once they are integrated in frequency (with the Bose function), give approximate Lorentzian shapes

with broader tails far away from q_{CDF} (or exactly Lorentzian if the CDF are in the classical regime $\Delta < T$).

Moreover these fits are done after subtracting the (H,H) scans from the (H,0) scans. This would be appropriate if the (H, 0) scans is on top of a background that is isotropic and equal to that from the (H,H) scan. However, the authors claim that the HH scans contain CDF and are thus not completely “background”. These inconsistencies provide for a somewhat confusing narrative. More significantly, one worries that these various subtractions and fits are somehow yielding incorrect results.

In principle, the referee is right in affirming that, by subtracting the HH from the H0 scan we are also partly removing a CDF contribution. However, this contribution is small and flat, and it does not modify the FWHM or the intensity of the peak, as the referee can check by comparing Fig. 2f to Supplementary Fig. 1a. We underline here that the curves obtained by this subtraction are used mostly to determine the FWHM in the (H,0) direction and the peak height at the q_{CDF} ; the isotropic part of the CDF, important for the strange metal behavior, is not relevant for the evaluation of the shape of the peak. To clarify this point, we have added the following sentence in the “Temperature dependence of CDF” section of the revised manuscript: << This subtraction allows us to remove from the quasi-elastic RIXS intensity the contribution of the elastic scattering due to surface defects, which is independent of the modulus of q ; at the same time, along the (H,H) direction the CDF contribution is still present though rather flat, so the shape, i.e. the FWHM, of the CDF peak, centered along the (H,0) direction, does not get modified by the subtraction procedure.>>

3. The main conclusions are only achieved after extensive multi-peak fitting of the energy and Q dependence to the high resolution data. However, the key “peak” of interest, the CDF peak, is the least resolved in raw data, being sandwiched in energy between elastic scattering and phonon modes, and also overlapping in energy with a paramagnon peak. As with this sort of fitting, the error bars on fitted peak parameters often do not provide a good representation of the certainty of the results. That is because there are often a range of parameter space that has similar reduced χ^2 values. Accordingly, a different choice of fitting procedure, such as using eqn. 1 instead of a Gaussian, or Voigt functions instead of Gaussians, or adding a bond-buckling phonon into the mix, would likely alter important parameters extracted from these fits and their doping dependence. While some of this may be inconsequential for the main story of the paper, it is not unreasonable to think that variation in fitting procedure could affect a key parameter, such as Δ , the energy of the CDF peak.

Before replying on the reliability of the fitting procedures, it is worth mentioning that the “main conclusions” of the paper rely not only on the determination of the CDF energy, but also on the temperature dependence of this peak and on the doping dependence of the CDF intensity. These dependencies are both obtained from the quasi-elastic integral of the medium energy resolution RIXS spectra and are evident even before doing any sort of fitting.

For what is of the determination of the CDF energy, it is correct what pointed out by the referee, i.e., that the energy of the CDF peak, sandwiched between the pure elastic and the breathing phonon peak, is known with a non-negligible error bar. We have done this fit in the best possible ways, i) considering Gaussian profiles to take the experimental energy resolution into account, and ii) trying to minimize the number of free parameters. In the latter case, adding more parameters, e.g., adding to the fit also a Gaussian profile deriving from the buckling phonons, would only apparently have resulted in a better fit. In particular, even though it is not

unplausible to think that buckling phonons could interfere with CDF, similarly to the higher-energy breathing phonons, we highlight here that the intensity vs q of the bond-buckling phonons is proportional to $\cos^2\pi q$. This implies that in the momentum range, $0.27 < H < 0.35$, where the CDF wave vector is present, the bucking phonon is quite weak, much smaller than the CDF contribution (which instead is maximum at q_{CDF}). To get an independent support to the results of the fitting, we have determined the energy values and/or the doping dependence of the energy by performing other type of analysis: i) the global fit of the H0 and HH scans with Eqs. 1 and 2, ii) the Lorentzian fit of the H0-HH peaks, iii) the fit vs T with the Bose occupation factor valid for q far from q_{CDF} . These procedures, based on medium energy resolution RIXS spectra that are different from the high-resolution spectra used for the direct determination of Δ , are all leading to consistent values for the energy vs doping, with a minimum at p^* . So, even though we are aware of the non-negligible error bar, we are confident of the robustness of the numbers we have extracted and about the conclusions we have reached.

4. The authors state that they correct all of the data for self-absorption corrections. However, the self-absorption corrections is different for a thin film vs a bulk single crystal. Moreover, these corrections can be substantial depending on the geometry, possibly introducing a systematic error that dominates the q dependence of the scattering. Similar to concern #3 above, systematic errors in this procedure can impact quantities such as the values of the FWHM of a peak or the peak position. Can the authors provide more details of the absorption correction? Was this done considering thin films in the case of YBCO and bulk crystals in the case of Bi2212.

We agree with the referee on the importance of the self-absorption corrections and on the risks of introducing artefacts in the data by this procedure. We therefore thank the referee for pointing out the need of a better presentation of this technical issue.

In brief, since one of the parameters in the formulas we have used to apply the correction is the sample thickness d , we have used a different self-absorption correction for the YBCO thin films and for the Bi2212 single crystal (which is approximated to a sample of infinite thickness). Moreover, the self-absorption correction is irrelevant to the determination of the FWHM and the H_{CDF} of the CDF peak. This is mainly because we have determined these two quantities from the difference between the H0 and the HH scans, which is independent of the applied correction (indeed, at each q , both along the H0 and HH directions, the correction is identical). However, we also highlight that, for both thin films and single crystals, the self-absorption correction modifies very little the shape of the H0 scans where the CDF peak center is located: both H_{CDF} and the FWHM of the CDF peak, within the uncertainty of our measurements, could be extracted also considering the bare H0 scan, and their values would be identical if extracted considering the H0 scans before or after the self-absorption correction.

In the revised version, we concisely but thoroughly discuss a better presentation of the procedure we have followed in the “RIXS measurements” paragraphs of the Methods section, while referring to our previous publications [Phys. Rev. Lett. 114, 217003 (2016); Phys. Rev. B 99, 134517 (2019)] for more details. Moreover, we have added in the supplementary the figure n° 8, comparing $H0$ -scans with and without the corrections, that shows the robustness of the results with respect to the self-absorption correction.

For convenience, we report here the text added to the Methods section.

“The RIXS spectra in the manuscript have been first corrected for self-absorption and then normalized to the integral of the inter-orbital dd excitations, in the range [-3 eV, -1 eV]. The self-absorption correction compensates for the possible reabsorption of scattered photons along

their path out of the sample from the scattering point. The probability of self-absorption depends on the absorption coefficients of the incident and scattered photons (which depend on their energy and polarization) and on the scattering geometry. It also depends on the total thickness of the sample. We have implemented the exact form of the correction that takes into account all the parameters: θ and χ , angles between the normal to the sample surface and the incident and emitted photon propagation direction; $\alpha_{T1,2}$ and $\alpha_{R1,2}$, total and resonant part of absorption coefficient for the incident/scattered photons, with $\alpha_T = \alpha_0 + \alpha_R$ and α_0 non-resonant pre-edge absorption coefficient not contributing to the RIXS signal; d , sample thickness. The quality of the correction depends on the knowledge of the absorption coefficients, which critically depend on the photon energy and polarization in the proximity of the absorption resonance. Conversely, the angular dependence is univocally known. The absorption coefficients were evaluated from the XAS spectra measured on every sample during each RIXS experiment. Their relative values are sufficient for the correction in case of thick samples (as single crystals, whose thickness can be approximated to infinite), whereas the absolute value of α_{T1} is needed for thin films, but it can be estimated from tabulated data of α_0 .

For clarity, we report here the self-absorption correction formulas. Calling $\eta_1(h\nu_2)$ the ideal RIXS spectrum measured with incident photon energy $h\nu_1$ and polarization ε_1 , the self-absorption modifies the spectral intensity point by point leading to:

$$I_1(h\nu_2) = \frac{\alpha_{R1}}{\alpha_{T1}} C_1(h\nu_2, \varepsilon_2) D_1(h\nu_2, \varepsilon_2) \eta_1(h\nu_2)$$

with

$$C_1\left(\theta, \chi, \frac{\alpha_{T2}(h\nu_2, \varepsilon_2)}{\alpha_{T1}}\right) = \left(1 + \frac{\alpha_{T2}(h\nu_2, \varepsilon_2) \sin \theta}{\alpha_{T1} \sin \chi}\right)^{-1}$$

$$D_1\left(\theta, \chi, \frac{\alpha_{T2}(h\nu_2, \varepsilon_2)}{\alpha_{T1}}, \alpha_{T1}, d\right) = 1 - \exp\left[-C_1 \cdot \alpha_{T1} \cdot \frac{d}{\sin \theta}\right]$$

We notice that for thick samples, $d \rightarrow \infty$ and $D_1 = 1$. Therefore, the spectrum $\eta_1(h\nu_2)$ that ideally would be measured in the absence of self-absorption from a thick sample is derived from the measured one $I_1(h\nu_2)$ as

$$\eta_1(h\nu_2) = \frac{\alpha_{T1}}{C_1 \cdot D_1} I_1(h\nu_2)$$

Thanks to this procedure we can safely compare the shape and intensity of spectra measured at different θ and χ angles (i.e., at different \mathbf{q} points) for a given sample and excitation energy (fixed absorption coefficients, also when the temperature is varied). Spectra of different samples can also be compared, although with some extra precautions due to the uncertainties related to the α coefficients that are not identical in different samples.

Reviewer #3 (Remarks to the Author):

This is an interesting piece of work. [...] The possible interrelations among pseudogap, charge/spin density fluctuations, Gaussian superconducting fluctuations (Nernst effect) and superconductivity are not clear yet. Against this backdrop, the present work is important. The experimental results and the analysis seems sound. I am in favor of publication of this paper. But there are few issues that require further clarification before this manuscript (m/s) can be accepted.

We thank the referee for the positive assessment of our paper, and we hope he/she will be satisfied by our revised version.

1. There are evidences that the CDF and PG are not related directly (Phys. Rev. Lett. 94, 237002 (2005); Physica C 387, 365 – 372 (2003); Supercond. Sci. Technol. 21 105017 (2008); Supercond. Sci. Technol. 21 125020 (2008)). At the same time, the PG vanishes at $p \sim 0.19$ (phys. stat. sol. (b) 215, 531 (1999); Physica C 349, 53 (2001); Phys. Rev. B 71, 054502 (2005); Scientific Reports volume 9, Article number: 14856 (2019)). This is the same hole content reported in this paper. It cannot be a coincidence. The authors need to address this issue.

We highlight that, as pointed out by the referee by putting forward those references, in the literature there is evidence that *charge density waves*, i.e., the quasi-static, long-ranged, charge modulations, are not directly related to the pseudogap. The newly discovered dynamical charge density fluctuations, on which our paper is based, still represent instead a piece in the puzzle of the cuprates whose understanding and connection with the other symmetry breaking orders and phases populating the HTS phase diagram has to be properly elucidated. And our current paper is one of the first ones, trying to shed light on this new feature.

Having said this, it is indeed remarkable – as pointed out by the referee - that the characteristic CDF energy (see Fig. 4d in the revised version) is in good quantitative agreement with the T^* line observed in the underdoped regime, up to the doping level p^* where it vanishes. However, our experiment cannot unravel the possible cause/effect hierarchy between charge density/quantum fluctuations and the pseudogap, or the relation of the pseudogap with the quantum criticality. Conversely, we have shown that at doping levels above p^* ($p=0.22$), i.e., outside the pseudogap region, the charge density fluctuations are still present, and their energy is still in quantitative agreement with the temperature where the strange metal ends. This last observation might bring to exclude a possible connection between CDF and the pseudogap, while supporting a CDF connection with the strange metal behavior.

However, further studies are needed to properly evaluate a possible link between these two entities. In the revised version, at the end of the “Discussion” section, we have added a new paragraph, all based on the possible connection between pseudogap and CDF where, among the other, we explicitly mention the coincidence of the $\Delta(p)$ and the $T^*(p)$ lines.

2. Charge ordering leads to the 1/8th anomaly which degrades the superconducting transition temperature. The hole content dependent PG energy shows featureless behavior close to the 1/8th doping (Phys. Rev. B 71, 054502 (2005); Supercond. Sci. Technol. 21 105017 (2008); Supercond. Sci. Technol. 21 125020 (2008)). A CDF analysis close to $p \sim 0.125$ will be informative. The authors should look at this matter.

We agree with the referee that the doping level $p=1/8$ is special for charge ordering, since the CDW peak exhibits there its strongest intensity and longest correlation length [Science 337, 821 (2012); Phys. Rev. B 90, 054513 (2014)]. And that this occurrence calls for an

investigation of the CDF peak at the same doping level. This is actually what we have done already in the previous version of the manuscript, and it is the main motivation behind the analysis we have performed to extract Ω , the energy of CDF far from the CDF wavevector \mathbf{q}_{CDF} . Indeed, since at the doping levels close to 1/8 the CDW is the strongest, when \mathbf{q} is close to $\mathbf{q}_{\text{CDW}} \approx \mathbf{q}_{\text{CDF}}$, the quasi-elastic intensity is dominated by the CDW and it becomes extremely difficult, within the energy resolution currently available with RIXS, to separate the CDF contribution from the CDW and to determine its energy Δ . Viceversa, at the same doping level, when we move very far from $\mathbf{q}_{\text{CDW}} \approx \mathbf{q}_{\text{CDF}}$, e.g. along the HH direction, the CDW contribution becomes negligible, being the CDW peak so narrow-in- \mathbf{q} , while the CDF intensity is still significant and its energy can be extracted. This is what we have presented in Fig. 4c: at doping levels close to 1/8, we cannot measure Δ but we can measure Ω . And the result is that, differently than for CDW, the $\Delta(\mathbf{p})$ line of CDF shows a featureless behavior close to 1/8 doping, which resembles the T^* line. In the revised version, as already mentioned in the reply to point #1 of the same referee, we have added a paragraph in the “Discussion” section to underline the similarity between these two lines.

3. In Fig. 4d, the $\Delta(\mathbf{p})$ line looks very similar to the PG(\mathbf{p}) line (Phys. Rev. B 71, 054502 (2005); Physica C 387, 365 – 372 (2003); Physica C 476, 10 (2016)). This resemblance needs addressing. It should be noted that the PG(\mathbf{p}) line may not be associated with any symmetry breaking (distinct phase) in the thermodynamic sense.

As already mentioned in the reply to point #1 of the same referee, we have addressed this resemblance in a new paragraph, that we have added at the end of the “Discussion” section. Moreover, while discussing the possible connection between CDF and pseudogap, we have added in the revised version references 53 to 57, most of them selected from the list mentioned by the referee.

Reviewers' comments:

Reviewer #1 (Remarks to the Author):

Arpaia et al. 2nd version.

The authors have significantly improved their manuscript with additional data and an extended discussion.

I am in favor of publication of the paper but there is one particular point that perplexes me in the new version.

The authors draw a connection with a neutron scattering work in LSCO (ref. 46). I see two problems here. First, LSCO has both spin and charge stripe orders, which is very different from YBCO that has only charge order. So, it sounds a bit like comparing apples and oranges here. Second, it does not seem to me that the authors of ref. 46 claim quantum criticality at p^* . They only observe that the characteristic energy of spin fluctuations at $p=0.22$ in LSCO (above $p^*=0.19$) is smaller than at $p=0.14$ in a proportion that is able to explain the enhancement of the specific heat at $p=0.22$. Earlier neutron scattering studies for $p=0.14$ found evidence of quantum critical behavior likely associated with the proximate spin stripe phase at $p<0.14$, not at p^* .

The comparison drawn by the authors is certainly relevant but it is rather misleading without further explanation and anyhow I believe that, as a matter of principle, works on LSCO cannot be used to claim universal mixed spin+charge quantum criticality in the cuprates. Furthermore, the situation in LSCO is unclear as the doping end of charge-stripe order and the evolution of charge-density fluctuations with doping are apparently not known. If CDW order ends above p^* , it is unlikely that the CDF energy drops at p^* as in YBCO. Then, the putative connection with the neutron scattering work (ref. 46) and with quantum criticality at p^* would not hold.

I would therefore advise revising this part of the discussion. The authors could simply state that spin properties of LSCO appear compatible with similar quantum criticality near p^* and they could add a reference such as <https://journals.aps.org/prb/abstract/10.1103/PhysRevB.106.054522> to complement ref. 46 on the complicated issues of magnetic and charge criticalities at p^* in LSCO.

Reviewer #2 (Remarks to the Author):

Unfortunately, despite efforts to address my concerns and many positive revisions to the manuscript, unfortunately, my primary concerns remain in the revised paper.

The analysis of the measurements involve to many assumptions regarding background subtraction and perhaps more importantly concern about the validity of parameters extracted from the multi-peak fitting. In my view, this uncertainty is not adequately addressed in the paper and limits the ability of readers to draw firm, unbiased conclusions from the paper. Accordingly, I do not support publication in Nature Communications.

Reviewer #3 (Remarks to the Author):

The modifications/clarifications made by the authors have significantly enhanced the quality of the paper. The revised version can be accepted for publication.

In support of the appeal after Editorial decision about manuscript NCOMMS-22-33778A.

Reviewers' comments:

Reviewer #1 (Remarks to the Author):

Arpaia et al. 2nd version.

The authors have significantly improved their manuscript with additional data and an extended discussion.

I am in favor of publication of the paper but there is one particular point that perplexes me in the new version.

The authors draw a connection with a neutron scattering work in LSCO (ref. 46). I see two problems here. First, LSCO has both spin and charge stripe orders, which is very different from YBCO that has only charge order. So, it sounds a bit like comparing apples and oranges here. Second, it does not seem to me that the authors of ref. 46 claim quantum criticality at p^* . They only observe that the characteristic energy of spin fluctuations at $p=0.22$ in LSCO (above $p^*=0.19$) is smaller than at $p=0.14$ in a proportion that is able to explain the enhancement of the specific heat at $p=0.22$. Earlier neutron scattering studies for $p=0.14$ found evidence of quantum critical behavior likely associated with the proximate spin stripe phase at $p<0.14$, not at p^* .

The comparison drawn by the authors is certainly relevant but it is rather misleading without further explanation and anyhow I believe that, as a matter of principle, works on LSCO cannot be used to claim universal mixed spin+charge quantum criticality in the cuprates. Furthermore, the situation in LSCO is unclear as the doping end of charge-stripe order and the evolution of charge-density fluctuations with doping are apparently not known. If CDW order ends above p^* , it is unlikely that the CDF energy drops at p^* as in YBCO. Then, the putative connection with the neutron scattering work (ref. 46) and with quantum criticality at p^* would not hold.

I would therefore advise revising this part of the discussion. The authors could simply state that spin properties of LSCO appear compatible with similar quantum criticality near p^* and they could add a reference such as <https://journals.aps.org/prb/abstract/10.1103/PhysRevB.106.054522> to complement ref. 46 on the complicated issues of magnetic and charge criticalities at p^* in LSCO.

We thank the referee for helping to better clarify this point. The comparison between our results and those of ref. 46 added in the resubmitted manuscript, was stimulated by two events. 1) after posting our manuscript on the arXiv, the authors of ref. 46 pointed to us the analogy between their inelastic neutron scattering data and our results; 2) a RIXS work showed for LSCO a phase diagram strongly resembling that of YBCO, with CDF pervading the whole normal state (npj Quantum Materials 8, 7 (2023)). So we would like to keep this reference. To avoid confusion and to take into consideration the other recent NMR work suggested by the referee also supporting a quantum criticality at p^* , we have however modified the sentence, which – in the new revised version – reads as: *“At our present knowledge, we cannot exclude a subdominant spin nature of the quantum fluctuations we have singled out with RIXS. However, we highlight here that recent inelastic neutron scattering and nuclear magnetic resonance results support a quantum criticality at p^* in $La_{2-x}Sr_xCuO_4$ due to spin excitations^{46,47}.”*

Reviewer #2 (Remarks to the Author):

Unfortunately, despite efforts to address my concerns and many positive revisions to the manuscript, unfortunately, my primary concerns remain in the revised paper.

The analysis of the measurements involve to many assumptions regarding background subtraction and perhaps more importantly concern about the validity of parameters extracted from the multi-peak fitting. In my view, this uncertainty is not adequately addressed in the paper and limits the ability of readers to draw firm, unbiased conclusions from the paper. Accordingly, I do not support publication in Nature Communications.

In order to refute the Referee's concerns, we briefly recap the experimental basis of our paper.

- i. The energy of the CDF peak is minimum at low temperature and $p=p^*$, and increases when increasing T and moving p away from p^* .
- ii. The intensity of the CDF peak is the highest at low temperature and at $p=p^*$, and decreases when increasing T and moving p away from p^* .

A crucial point is therefore how we have determined the energy and the intensity of the CDF peak. In the following, we list the different procedures and strategies we have followed (underlined, we have put the analysis which have been criticized by the referee):

Energy:

- a) Multipeak fit of the high-resolution (HiRes) spectra (we extract only one parameter, the center of the Gaussian of the CDF peak)
- b) FWHM of the HO-HH peaks (integrated quasi-elastic intensity vs q , medium-resolution (MedRes) spectra).
- c) Global fit of the HO and of the HH scans with theory (integrated quasi-elastic intensity vs q , medium resolution spectra)
- d) Fit of the HH scans with Bose (integrated quasi-elastic intensity vs T , medium resolution spectra)

Intensity:

- a) Height of the HO-HH peaks (integrated quasi-elastic intensity vs q , medium resolution spectra).

We can now discuss the two criticisms by the Referee.

1) Multipeak fitting. We were aware from the beginning that the energy scale of CDF (10-30 meV) is at the limit of even the best RIXS instruments available today. Therefore, we combined **four** different ways of evaluating the CDF energy, not relying only on the multi-peak fit of high-resolution (HiRes) RIXS spectra. Still, the "HiRes fit", i.e. the fit of the HiRes spectra, shown in Fig.1 of the paper, represents the most direct way to measure the energy and deserved to be done. In this case, the CDF energy, the parameter of our interest, is given by the center of the Gaussian representing the CDF, which is between the elastic peak and the breathing phonon peak. We use a Gaussian line shape because the peak is clearly resolution limited. This is therefore the only parameter of the HiRes fit, relevant for the conclusions of our paper. The validity of this procedure and of the extracted parameter is supported by several facts:

- i) The CDF energy obtained from the HiRes fitting is consistent with that determined using the other 3 approaches
- ii) The energies of the breathing phonons, obtained as free parameters from the HiRes fit, result in full agreement at all q with inelastic neutron scattering results, as we have highlighted in the caption of Figure 1. A significant error in the determination of the CDF energy would directly affect the position of the energy of the breathing mode.

iii) The value of the CDF energy, on which we base the conclusions of our paper, is the one at $q=q_{\text{CDF}}$, which is in the range between 0.28 rlu and 0.35 rlu, depending on p . There the intensity of the breathing phonon peak, going as $\sin^2(\pi q)$, and therefore being maximum at $q=0.50$, is still rather weak (as we have also emphasized in the methods, subsection “Alignment and fit of the high resolution RIXS spectra”). Moreover, the elastic peak is maximum at $\Gamma=(0,0)$ and decreases as q increases (as shown in Supplementary Fig. 6a). The consequence is that the CDF energy value of largest interest, i.e. at $q=q_{\text{CDF}}$, is the one estimated with a highest certainty, because the intensity of the two adjacent peaks is relatively weak.

2) Background subtraction. In brief, the reason for fitting the HO scans, after subtraction of the background measured along the HH direction, is to minimize some arbitrariness intrinsic of the analysis and fit of the CDW/CDF peaks commonly done in literature, and to be conceptually more correct. However, the result of the fit is unchanged if one analyses the HO scans directly or after subtraction of the HH scans. We explain this in detail in the following.

Already in the first, pioneering, RIXS (RIXS and energy integrated RIXS) experiment, by some of us, showing the presence of CDW in a cuprate system (YBCO), the temperature dependence of the CDW peak was thoroughly investigated (see Fig. L1a-b).

Figure L1

The way to compare the measurements at different temperatures (and therefore extracting information as the temperature dependence of the CDW), was the following. First, the q -scans measured at different temperatures were vertically translated, aligning the tails of the peaks, which are almost no temperature dependent. After that, the q scans at different temperatures were subtracted by a reference scan measured at the threshold (high) temperature, above which the scans appear unchanged. This procedure, which became a common habit in the field, used by a series of important articles by several groups (see Fig. L1c-d-e-f), overlooks the raw intensities, which are changing at any q as the temperature varies (see for instance Fig. S1a-b of our manuscript) and intrinsically cancels out any charge order signal possibly present at and above the threshold temperature. In 2019 we discovered that the charge order phenomenon is broader than commonly

thought, and we reported in a Science paper (365, 906 (2019)) the evidence of charge density fluctuations (CDF), which are represented by the weakly T -dependent signal, centered along the H0 direction, previously disregarded and used as background. The important question arising is: which is the correct background under the CDF peak? A background is indeed needed to remove from the quasi-elastic intensity the contribution of the elastic scattering due to surface defects, which is centered at Γ (fortunately, this is quite large only at low q values, i.e. rather far from q_{CDF}). Since this contribution is independent of the modulus of q , we have chosen to use the HH scans. As we have already explicitly written in the main text, even though along the HH direction the CDF contribution is still present, it is rather flat: *the shape, i.e. the FWHM and the difference between the intensity on the maximum and that on the tails, of the CDF peak is not altered by the subtraction procedure*. This is a crucial point, we ask the referee to not overlook. In the case of YBCO, we can for instance compare Fig. 2f and Fig. S1a, reported below. Since the background is almost flat (see Fig. S1b, also reported below), *the peak properties extracted from the peak, ie the FWHM - therefore the energy - and the intensity, are independent of the background (HH) subtraction*. Figures 2 and S1a-b also show that the H0-HH subtraction automatically cancel out the isotropic increase of signal, following the Bose statistics, and brings in a natural way to the overlapped tails in the temperature dependence, which previously were achieved by arbitrary vertical translation of the data.

Fig. 2

Fig. S1

To make our latest statement more quantitative and to neutralize the concerns of Referee #2 about the “assumptions regarding background subtraction”, we have calculated, in the case of YBCO $p=0.185$, both the FWHM and the intensity of the CDW peak in a more traditional manner, i.e. directly from the H0 scans of Fig. S1a. These scans, after translation of the integrated intensities to overlap the tails at different temperatures, look like in Fig. L2 (see similarities with the H0-HH peaks in Fig. 2):

Figure L2

We have fitted these scans with Lorentzian curves, similarly as we have done in the paper to the scans in Fig. 2. For each scan we have calculated the height and the FWHM of the Lorentzian, and we have plotted in Fig. L3c-d these two parameters as a function of the temperature:

Figure L3

For comparison, in Fig. L3a-b we have instead plotted the results of the fit done on the $H0$ - HH curves, as in Fig. 2g-h of the manuscript. *The results (both for the FWHM and for the height) are nearly identical.* This unambiguously shows that our background subtraction does not affect the values we have extracted in the paper for the energy (determined from the FWHM) and for the intensity of the CDF peak.

In conclusion, we present again the list shown in the beginning of this reply letter, including the different procedures and strategies we have followed to extract the energy and the intensity values of CDF. The difference here is that we can substitute the analysis of the $H0$ - HH peaks – which were casting doubts in the referee - with those on the $H0$ peaks, since we have just demonstrated they are identical.

Energy:

- Multipeak fit of the high-resolution spectra (we extract only one parameter, the center of the gaussian)
- FWHM of the $H0$ peaks (integrated quasi-elastic intensity vs q , medium-resolution spectra).
- Global fit of the $H0$ and of the HH scans with theory (integrated quasi-elastic intensity vs q , medium resolution spectra)
- Fit of the HH scans with Bose (integrated quasi-elastic intensity vs T , medium resolution spectra)

Intensity:

- Height of the $H0$ peaks (integrated quasi-elastic intensity vs q , medium resolution spectra).

We recognize that some residual uncertainty might remain be about the multipeak fit, underlined for clarity. However, we reiterate that that uncertainty is removed by the fact that other procedures lead to consistent results on the CDF energy, as explained above.

REVIEWER COMMENTS

Reviewer #2 (Remarks to the Author):

I appreciate the authors providing additional information regarding fitting of (H,0) scans. However, the additional analysis does not alleviate my primary concerns. I ultimately do not think it is possible to extract the FWHM of these peaks from the present data in a manner that provides details such as the temperature dependence of the FWHM with sufficient accuracy to draw significant conclusions from. The systematic uncertainties exceed the statistical error bars presented in many of the plots and figures. The peaks are too broad relative to the range of Q measured and the background too large to yield results that I would rely on. In other studies, this may be fine because the central conclusions of the paper may not depend on difficult to extract details such as the T dependence of the FWHM. However that is not the case here.

Similarly, I remain unconvinced that important parameters from the multiplex fitting of the high res RIXS measurements can be deduced from the present data without systematic errors.

I would certainly grant that the authors have shown that their data is consistent with the models for charge density fluctuations presented in the paper. However, I do not recommend publication in Nature Communications. It is my view that publication in a high impact journal such as Nature Communications requires a more definitive set of experiments.

Reviewer #4 (Remarks to the Author):

In response to the Editor's request to comment on the exchange between Reviewer #2 and the authors, I have thoroughly examined Reviewer #2's comments. I agree with Reviewer #2's primary concern regarding the validity of the parameters derived from the multi-peak fitting.

Reviewer #2 has raised two specific concerns regarding the data analysis. Firstly, they questioned the assumptions made in the background subtraction process. Secondly, they expressed doubts about the validity of the curve fitting model employed, which incorporates four components: elastic peak, charge density fluctuation (CDF), bond-stretching phonon, and paramagnon. However, the fitting

scheme of the RIXS spectra presented in the manuscript, such as in Fig. 1c, appears to be questionable. As described in the Methods section, it is now widely accepted that a typical high-resolution RIXS spectrum of doped cuprates comprises contributions from bond-buckling phonons and bond-stretching phonons (see Refs. 41, 68, and 69). The accurate determination of the CDF energy critically depends on the number of phonon components considered. Notably, Ref. 41 demonstrates that O K-edge RIXS spectra of LSCO, recorded with an enhanced energy resolution of 16 meV, exhibit the presence of buckling phonons. Since the determination of the CDF energy and related parameters is pivotal to the conclusions drawn in this paper, the absence of buckling phonons would substantially impact the CDF energy determination. Hence, Reviewer #2's critique is valid. Regarding the background subtraction approach using the fitting results of the HH scan, the method utilized in the manuscript is reasonable. However, it is worth noting that such background subtraction may be unnecessary if the RIXS energy resolution is improved, as demonstrated in Ref. 41.

REVIEWER COMMENTS

Reviewer #4 (Remarks to the Author):

In response to the Editor's request to comment on the exchange between Reviewer #2 and the authors, I have thoroughly examined Reviewer #2's comments. I agree with Reviewer #2's primary concern regarding the validity of the parameters derived from the multi-peak fitting.

Reviewer #2 has raised two specific concerns regarding the data analysis. Firstly, they questioned the assumptions made in the background subtraction process. Secondly, they expressed doubts about the validity of the curve fitting model employed, which incorporates four components: elastic peak, charge density fluctuation (CDF), bond-stretching phonon, and paramagnon. However, the fitting scheme of the RIXS spectra presented in the manuscript, such as in Fig. 1c, appears to be questionable. As described in the Methods section, it is now widely accepted that a typical high-resolution RIXS spectrum of doped cuprates comprises contributions from bond-buckling phonons and bond-stretching phonons (see Refs. 41, 68, and 69). The accurate determination of the CDF energy critically depends on the number of phonon components considered. Notably, Ref. 41 demonstrates that O K-edge RIXS spectra of LSCO, recorded with an enhanced energy resolution of 16 meV, exhibit the presence of buckling phonons. Since the determination of the CDF energy and related parameters is pivotal to the conclusions drawn in this paper, the absence of buckling phonons would substantially impact the CDF energy determination. Hence, Reviewer #2's critique is valid. Regarding the background subtraction approach using the fitting results of the HH scan, the method utilized in the manuscript is reasonable. However, it is worth noting that such background subtraction may be unnecessary if the RIXS energy resolution is improved, as demonstrated in Ref. 41.

We thank the referee for taking the time to carefully read our manuscript and to evaluate the scientific exchange between us and Reviewer #2. Their comments offer us the opportunity i) to justify our choice to measure at the Cu L₃ edge, ii) to assess the solidity of the position in energy of the CDF peak as extracted from the multiplex fit, and finally iii) to clarify the relation between the CDF and the buckling phonons, which was already formerly discussed in the methods section.

- ✦ As regards point i), we agree with the referee that RIXS spectra can have better resolution when measured at the O K edge, as done in Ref. 41 to study with finer details the phonons of LSCO. Indeed, at the Cu L₃ edge it is extremely hard to do better than we show in Figures 1, 3a-e and S5, i.e., to measure with a resolution better than 40 meV a complete map vs q, at different temperatures, with sufficient statistical quality to make a fit. However, at Cu L₃ the count rate is higher than at O K and the whole Brillouin zone can be mapped, two factors that are important for our systematic work across the phase diagram. Moreover, cross sections for phonons are not identical at the two edges, and this has to be kept in mind when comparing different datasets and their analysis. The origin of this discrepancy lies in the different nature of the intermediate state: at the Cu L₃ edge, the excited electron is more strongly bound to the core hole in comparison to the O K edge. Consequently, the relative intensity *of the different phonons, including the buckling and the breathing branches, are therefore different at the Cu L₃ and at the O K edge*. Some of these important differences emerge in the following papers [Phys. Rev. Lett. 123, 027001 (2019); Phys. Rev. Res. 2, 023231 (2020)], and [Phys. Rev. B 105, 115105 (2022)]. In particular, it emerges that *at the O K edge, the intensity of the buckling phonons is particularly strong*. By considering the $\cos^2(\pi q)$ dependence of the intensity, this implies that at low momenta, the buckling phonon provides the main spectral contribution to the RIXS spectra below 100 meV [PNAS 117, 16219 (2020); Phys. Rev. B 105,

115105 (2022)]. At momenta values close to $q_{\text{CDW/CDF}}$, as $q=0.25$, the intensity of the buckling phonons is still strong enough to be higher than that of the breathing phonons (whose intensity at the opposite is characterized by a $\sin^2(\pi q)$ dependence), and to allow distinguishing both the A_{1g} and the B_{1g} modes, as demonstrated in Ref. 41. A fit of the low energy region of the spectra at the O K edge cannot therefore ignore the buckling modes. On the contrary at the Cu L_3 edge, the intensity of the buckling phonons is much weaker than that of the breathing mode. In [Phys. Rev. X 6, 041019 (2016)] it is shown that at the Cu L_3 edge, the intensity of the B_{1g} modes (having the lowest energy among the buckling branches) is negligible; moreover, the intensity of the higher energy A_{1g} mode is weak enough to be comparable to that of the breathing phonons already at low q , where the intensity of the buckling phonons is the highest and the intensity of the breathing phonons is the smallest. The figure below, reproduced after Figure 9 of the aforementioned PRX, demonstrates that the intensity is mainly given at very low energy by the acoustic phonons, which are strongly coupled to the CDF [Ref 41], and at high energy by the breathing phonons. The buckling phonon, at intermediate energy, is weaker. Therefore, it is acceptable to ignore the buckling phonons in our fittings for the determination of the CDF energy. This approximation would not be acceptable at the O K edge. This has been confirmed by a detailed analysis of our data, which is presented below.

Figure R1: Sum of all the contributions from phonons to RIXS at the Cu L_3 edge. Momentum transfer is given in units of π/a . [from PRX 6, 041019 (2016)].

- As regards point ii), in our manuscript we have performed the multi-peak fit of the high-resolution spectra assuming only three peaks (pure elastic, CDF, breathing phonons, in addition to the phonon overtones and to the paramagnons, whose main contribution is however at energies higher than 100 meV). This choice allowed us to reduce the number of free parameters and get more significant results. Indeed, adding to the fit also a Gaussian profile for the buckling phonons would only apparently lead to a better fit. As explained in the previous paragraph, the intensity of the buckling phonons at the Cu L_3 edge is in general very weak and, being proportional to $\cos^2\pi q$, is even weaker in the momentum range, $0.27 < q < 0.35$ (in r.l.u.), where the CDF contribution is the strongest and its energy Δ is defined. However, stimulated by the referee's comments, we performed a new multi-peak fit of the high-energy resolution spectra. Here, we have included a fourth contribution, taking into account the buckling phonons, given by a resolution-limited Gaussian, centered at intermediate energies between the CDF Gaussian and the breathing phonon Gaussian. As starting point, we have confined the energy of this additional peak in the range between 35 and 40 meV, where inelastic neutron scattering, Raman and ARPES experiments have detected the buckling phonons in Bi2212 and YBCO [Phys. Rev. Lett. 93, 117003 (2004); Phys. Stat. Sol. (b) 242, 11 (2005); Phys. Rev. B 78, 132503 (2008)]. We have selected, for each sample, the lowest value of q among the measured spectra. The result is summarized in Figure R2, where we have shown, for the slightly overdoped ($p=0.19$) YBCO and Bi2212 the spectrum respectively measured at $q=0.15$ rlu and at $q=0.08$ rlu. We find that the intensity of the buckling phonons (black Gaussian)

is comparable with that of the breathing phonons (orange Gaussian), in agreement with the calculations shown in Fig. R1, and much smaller than that of the CDF (red Gaussian), even at a q value different from q_{CDF} . This very first test confirms the marginal role of buckling phonons in determining the total spectral weight of the quasi-elastic region of our spectra.

Figure R2: Multi-peak fit of the high resolution RIXS spectra (blue circles) we have investigated at the lowest q value for the slightly overdoped (a) YBCO and (b) Bi2212. Following the same color code used in the main text, the green, red, orange, and violet Gaussians and the region below the grey dashed line represent respectively the pure elastic, the CDF, the bond-stretching phonon modes, the phonon overtone, and the paramagnons. In addition, the black Gaussian represents the bond-buckling phonon modes at 35-40 meV. The sum of all the contributions is given by the blue curve.

After this preliminary test, we repeated the fitting to the whole range of q . We have performed the fit following two different strategies, trying in both cases to determine a sort of upper limit on the role the buckling phonons can have at each q , overestimating on purpose their intensity. In the first case, we have confined the buckling phonon energy in the range between 35 and 40 meV, fixing at all q the intensity to the value we have measured at the lowest q value. In the second case, we have kept free both the intensity and the energy position of the buckling phonons. In both cases, as we can see in Figures R3, R4 and R5 below, the intensity of the buckling phonons is relatively weak at any q . Moreover, in particular for $q \approx q_{\text{CDF}}$, it causes an energy shift of the CDF Gaussian, with respect to the fit without buckling phonons, smaller than 4 meV. We take this number as the uncertainty on the position of the energies in the high-resolution spectra (as already explained in the methods section). The results are summarized in Figure R6, for the $p=0.19$ Bi2212 sample, where we compare the area of the low-energy Gaussians either excluding (see Fig. R6(a)) or including (see Fig. R6(b,c)) the buckling phonon contribution. Here, the latter is obtained by keeping the intensity determined at the lowest q constant at all q (which, as already mentioned before, is an overestimation). The result is that the buckling contribution is tiny, comparable only to the breathing phonons at the lowest wavevectors, and it does not significantly modify the parameters (height and/or FWHM) of the other low-energy features.

In conclusion, we have experimentally confirmed in our spectra that the contribution of the buckling phonons is negligible at the Cu L3 edge, i.e., their spectral weight is low enough that the energy of the other low-energy features (CDF, breathing phonons, etc.) is substantially the same, within our experimental uncertainty, either considering or ignoring the buckling phonon contribution.

Figure R3: High resolution RIXS spectrum at $q=q_{\text{CDF}}=0.26$ rlu for the slightly overdoped ($p=0.19$) Bi2212. The multi-peak fit is performed (a) without considering the buckling phonons; (b) considering the buckling phonon energy fixed at 35 meV and the intensity as measured at $q = 0.08$ rlu; (c) keeping both the energy and the intensity of the buckling phonons free parameters

Figure R4: High resolution RIXS spectrum at $q \approx q_{\text{CDF}}=0.31$ rlu for the slightly overdoped ($p=0.19$) YBCO. The multi-peak fit is performed (a) without considering the buckling phonons; (b) considering the buckling phonon energy fixed at 40 meV and the intensity as measured at $q = 0.15$ rlu; (c) keeping both the energy and the intensity of the buckling phonons free parameters

Figure R5: High resolution RIXS spectrum at $q=q_{\text{CDF}}=0.35$ rlu for the strongly underdoped ($p=0.06$) YBCO. The multi-peak fit is performed (a) without considering the buckling phonons; (b) considering the buckling phonon energy fixed at 40 meV and the intensity as measured at $q = 0.07$ rlu; (c) keeping both the energy and the intensity of the buckling phonons free parameters

Figure R6: Area of the low-energy Gaussians, resulting from the multi-peak fit of the ($p=0.19$) Bi2212 spectra. Here, the buckling phonon energy has been fixed at 40 meV and the intensity at all q is determined at $q = 0.08$ rlu

✦ As regards point iii), in the methods section we have briefly discussed a possible relation between the CDF and the buckling phonons. This is rather natural to infer, taking into consideration that: 1) charge density waves and fluctuations induce an energy softening and intensity enhancement of the breathing phonons at $q \approx/\gtrsim q_{\text{CDW/CDF}}$ [Nat. Phys. 13, 952 (2017); Nat. Phys. 17, 53 (2021)], as also confirmed by the data of our manuscript; 2) at the O K edge, where their intensity is boosted, buckling phonons present the same anomalies [PNAS 117, 16219 (2020); Phys. Rev. X 11, 041038 (2021)]. At the Cu L₃ edge, the buckling phonons should therefore feel the interaction with charge order. However, as previously discussed, their intensity is so low that, within the energy resolution nowadays available, detecting any sign of anomaly is extremely problematic. The spectral weight associated with these anomalies represents a tiny fraction of the total CDF signal in the system: *even considering an entwining between CDF and buckling phonons, we exclude the CDF peak is significantly modified in intensity and FWHM (both in q and in energy) as a consequence of this interaction.* We have clarified this point in the new version of the methods section, where we have also highlighted that, as also pointed out in Ref. 41, *it is very plausible a coupling of the CDF with the acoustic phonons*, whose intensity is rather strong at the Cu L₃ edge (see Fig. R1) and whose energy is in the same range as CDF.

As a final remark, we highlight that the doubts of Referee #4 regarding the fitting scheme (as well as the concerns of Referee #2 regarding the analysis of the high-resolution spectra) are connected to the estimation of the absolute energy of CDF. *However, we iterate what we have written already in the first version of the manuscript, that the relative changes of the CDF energy and intensity versus temperature and doping are the main sign of the quantum criticality, driven by charge fluctuations.* These relative changes cannot be questioned because they go beyond the uncertainty of any of the analysis schemes we have used, since we have pointed out on purpose the attention to the two far extremes both of the doping (strongly underdoped and slightly overdoped) and of the temperature (T_c and T_{room}) range. The absolute value of the CDF energy enters only in the final part of our discussion where we make the link between the CDF and the strange metal phase of cuprates.

REVIEWERS' COMMENTS

Reviewer #4 (Remarks to the Author):

My previous remarks were prompted by the Editor's request to address Reviewer #2's concerns about the validity of the parameters employed in the multi-peak fitting scheme by Arpaia et al. Specifically, the energy resolution in current RIXS measurements significantly exceeds that of the CDF energy, and the RIXS spectrum encompasses various phonon modes, including acoustic, buckling, and breathing phonons. Moreover, the tail of the paramagnon feature extends to the spectral weight resulting from these phonon signals.

In their revised manuscript and response to my earlier comments, the authors comprehensively explain the distinctions between Cu L-edge and O K-edge RIXS. They have addressed the negligible contribution of buckling phonons in their curve-fitting analysis. While it is acknowledged that Cu L-edge RIXS contains signals of various phonon modes and exhibits minor buckling phonon contributions, the revised manuscript overlooks the role of acoustic phonons. As mentioned in the authors' response, the low-energy spectral weight in Cu L-edge RIXS is significantly influenced by acoustic phonons, whose energy is comparable to the CDF energy. It is imperative to include discussions regarding the role of acoustic phonons, as their energy is comparable to that of the CDF. The separation of phonon and CDF signals in analyzing their Cu L-edge RIXS remains unclear. In conclusion, doubts persist regarding the curve-fitting analysis presented in the revised manuscript.

REVIEWER COMMENTS

Reviewer #4 (Remarks to the Author):

My previous remarks were prompted by the Editor's request to address Reviewer #2's concerns about the validity of the parameters employed in the multi-peak fitting scheme by Arpaia et al. Specifically, the energy resolution in current RIXS measurements significantly exceeds that of the CDF energy, and the RIXS spectrum encompasses various phonon modes, including acoustic, buckling, and breathing phonons. Moreover, the tail of the paramagnon feature extends to the spectral weight resulting from these phonon signals.

In their revised manuscript and response to my earlier comments, the authors comprehensively explain the distinctions between Cu L-edge and O K-edge RIXS. They have addressed the negligible contribution of buckling phonons in their curve-fitting analysis. While it is acknowledged that Cu L-edge RIXS contains signals of various phonon modes and exhibits minor buckling phonon contributions, the revised manuscript overlooks the role of acoustic phonons. As mentioned in the authors' response, the low-energy spectral weight in Cu L-edge RIXS is significantly influenced by acoustic phonons, whose energy is comparable to the CDF energy. It is imperative to include discussions regarding the role of acoustic phonons, as their energy is comparable to that of the CDF. The separation of phonon and CDF signals in analyzing their Cu L-edge RIXS remains unclear. In conclusion, doubts persist regarding the curve-fitting analysis presented in the revised manuscript.

We thank the referee for their positive evaluation of our paper and for their valuable suggestions. In response to their comments, we have incorporated a concise discussion at the manuscript end, which is highlighted in blue in the revised version. There, we discuss the relationship between CDF and phonons, particularly the acoustic branch. As previously conveyed to the referee in our prior correspondence, we propose a strong coupling between CDF and acoustic phonons. This proposal is suggested by the similar energy ranges associated with CDF and acoustic phonons, and supported by a recent RIXS study at the O K edge (reference 38 in our revised manuscript), which was conducted with a 16 meV resolution.

Furthermore, we have made substantial revisions to the abstract to conform to the 150-word limit mandated by the editorial policy.